

# Fire, vegetation and Holocene climate in the south-eastern Tibetan Plateau: a multi-biomarker reconstruction from Paru Co

Alice Callegaro[1,2], Felipe Matsubara Pereira[1], Dario Battistel[l], Natalie M. Kehrwald[3], Broxton W. Bird[4], Torben Kirchgeorg[1] and Carlo Barbante[1,2]

[1]Dipartimento di Scienze Ambientali, Informatica e Statistica, Università Ca' Foscari Venezia, Venezia, 30172, Italia
[2]Istituto per la Dinamica dei Processi Ambientali-CNR, Venezia, 30172, Italia
[3]Geosciences and Environmental Climate Change Science Center, U.S. Geological Survey, Denver Federal Center, Lakewood, CO 80225, USA
[4]Department of Earth Sciences, Indiana University–Purdue University, Indianapolis, IN 46208, USA

*Correspondence to*: Alice Callegaro (alice.callegaro@unive.it) and Natalie M. Kehrwald (nkehrwald@usgs.gov)

**Abstract.** The fire history of the Tibetan Plateau over centennial to millennial timescales is still unknown. Recent ice core studies reconstruct fire history over the past few decades but do not extend through the Holocene. Lacustrine sedimentary cores, however, provide continuous records of large-scale and local environmental modifications due to their accumulation of

specific organic molecular markers throughout the past millennia. In order to reconstruct Holocene fire events and vegetation changes occurring on the south-eastern Tibetan Plateau and the surrounding areas, we improved and integrated previous analytical methods. The multi-proxy procedure was applied to samples retrieved from Paru Co, a small lake located in the Nyainqentanglha Mountains (29°47'45.6" N; 92°21'07.2" E; 4845 m a.s.l.). The investigated biomarkers include *n*-alkanes as indicators of vegetation, polycyclic aromatic hydrocarbons (PAHs) as combustion proxies, faecal sterols and stanols (FeSts)

as indicators of the presence of humans or grazing animals and finally monosaccharide anhydrides (MAs) as specific markers of vegetation burning processes. Relatively high concentrations of both MAs and PAHs demonstrate intense local biomass burning activity during the early Holocene (10.9-10.7 cal ky BP), which correspond to a drier climate following deglaciation. High concentrations of MAs but not PAHs between 10.7-9 cal ky BP suggest a period of regional biomass burning followed by a decreasing fire trend through the mid-late Holocene. This fire history is consistent with local vegetation changes

reconstructed from both *n*-alkanes and regional pollen records, where vegetation types depend on the centennial-scale intensity of monsoon precipitation. FeSts were below detection limits for most of the samples, suggesting limited direct human influences on fire regime and vegetation changes in the lake's catchment. Climate is the main influence on fire activity recorded in Paru Co over millennial timescales, where biomass burning fluctuates in response to alternating warm/humid and cool/dry periods.





## 1 Introduction

Terrestrial vegetation is the primary source of biomass burning (Simoneit et al., 1999), where this combustion is due to both natural processes and anthropogenic activities. Fire-related forest clearance contributes to the global burden of greenhouse gases and causes associated global warming (Bowman et al., 2009). The impacts of global climate change on the frequency,

intensity, duration, and location of biomass burning are not well known and the contribution of fire emissions to past and future atmospheric composition are not clear (IPCC, 2014). Therefore, past and present biomass burning needs to be characterized and accurately mapped in order to investigate interactions with weather, climate and landscape dynamics over a range of spatiotemporal scales.

Lake sediments archive high-resolution histories of sediment flux, as well as hydrological and ecological modifications, provided as long as the lakes persist and preserve sediments through time (Yan and Wünnemann, 2014). Numerous recent studies demonstrate climatic variations throughout China and surrounding areas during the Holocene using lacustrine sediment as archives of the past climate (Bird et al., 2017; Dietze et al., 2013; Liu et al., 2009; Opitz et al., 2012; Saini et al., 2017; Yanhong et al., 2006). The paleoclimate proxies used in these studies including carbonate percentages, mineralogy, grain-size

distribution, elemental geochemistry, stable isotope composition, leaf wax long-chain $n$-alkanes, aquatic diatoms and terrestrial pollens, collectively record changes in hydroclimate and ecosystem processes. Only a few studies examine past biomass burning by using charcoal in Tibetan sediments (Herrmann et al., 2010; Miao et al., 2017) or black carbon, levoglucosan and ammonia as combustion proxies in ice cores (Kaspari et al., 2011; Ming et al., 2008; Shugui et al., 2003; Xu et al., 2009; You et al., 2016b). However, these studies only investigate the fire history of the last century. Polycyclic aromatic hydrocarbons

(PAHs) are reported in the lake sediments from the Tibetan Plateau (TP) spanning the last two centuries (Yang et al., 2016). To the best of our knowledge, however, no studies examine PAHs or monosaccharide anhydrides in sediments from the TP for the entire Holocene.

The combination of innovative molecular markers helps infer past fires, vegetation and human interactions, as highlighted in

sediment cores analysis from Guatemala (Schüpbach et al., 2015) and East Africa (Battistel et al., 2016). Here, we use biomarkers that are produced in specific environmental conditions and then transported, accumulated, and stored in lacustrine sediments: monosaccharide anhydrides (MAs); fecal sterols and stanols (FeSts); polycyclic aromatic hydrocarbons (PAHs); and normal($n$)-alkanes. Significant concentrations of these compounds in buried sediments are present in soil and sedimentary archives with ages even older than 10 cal ky BP (D'Anjou et al., 2012; Johnsen et al., 2005; Schüpbach et al., 2015), thus

suggesting that degradation, if happening, is a low-kinetic process (Battistel et al., 2016) and that these compounds resist over the Holocene. MAs are specific tracers of vegetation combustion (Simoneit, 2002; Zangrando et al., 2013). Cellulose pyrolysis creates the molecular marker levoglucosan (1,6-anhydro-β-D-glucopyranose) (Simoneit et al., 1999), while hemicellulose combustion produces the isomers mannosan (1,6-anhydro-β-D-mannopyranose) and galactosan (1,6-anhydro-β-D-



galactopyranose) (Kuo et al., 2011). Several studies examine levoglucosan (L), mannosan (M) and galactosan (G) in aerosols and ice cores (Kehrwald, 2012; Simoneit, 2002; Yao et al., 2013; Zennaro et al., 2014; Zhang et al., 2008), as well as in sediment cores (Battistel et al., 2016; Kirchgeorg et al., 2014; Schüpbach et al., 2015), demonstrating the suitability of MAs as paleofire proxies. PAHs are a wide group of organic compounds made up of two or more benzene rings combined together

in linear, angular, or clustered arrangements (Zakir Hossain et al., 2013). The physical properties of PAHs, such as low aqueous solubility and high lipophilicity, prevent microbial utilization and promote their accumulation in the particulates in terrestrial environments (Johnsen et al., 2005). This class of molecules is produced by incomplete combustion during a wide range of natural and anthropogenic processes, such as volcanic eruptions, vegetation burning, fossil fuels, garbage and cigarette or car emissions (Abdel-shafy and Mansour, 2016; Kim et al., 2013; Lima et al., 2005). PAHs are semi-volatile, persistent and

ubiquitous in the environment, and therefore are commonly detected in soil, air, and water (Abdel-shafy and Mansour, 2016; Johnsen et al., 2005). However, due to their multiple possible sources, only a few studies consider PAHs as tracers of biomass burning in past climate archives such as sediments (Jiang et al., 1998) and ice (Gabrieli et al., 2010).

Leaf waxes are preserved in sediments and can help determine past vegetation in a lake catchment. The cuticular wax layer of

terrestrial plants consists predominantly of long-chain hydrocarbons and creates a protective barrier that helps maintain the plant's integrity within an intrinsically hostile environment (Sheperd and Griffiths, 2006). The leaf wax of higher plants is difficult to degrade during transport, deposition and burial (Cui et al., 2008). Different types of plants have diverse chain-lengths of $n$-alkanes (Diefendorf and Freimuth, 2017). Angiosperms generally produce more $n$-alkanes than gymnosperms; however, chain-length distributions are highly variable within plant groups, and especially for conifers where the Cupressaceae

group tends to have long chain $n$-alkanes, while the Pinaceae group tends to have relatively short chain $n$-alkanes (Diefendorf and Freimuth, 2017; Diefendorf et al., 2015). *Sphagnum* mosses are among the few plants that provide a characteristic signal as these mosses are marked by the predominance of $C_{23}$ and $C_{25}$ (Bush and McInerney, 2013). Long chain $n$-alkanes ($C_{27}$–$C_{33}$) with a strong odd/even predominance are usually interpreted to originate from terrestrial plants; mid-chain $n$-alkanes ($C_{20}$–$C_{25}$) are mainly present in aquatic macrophytes; bacteria, algae and fungi primarily produce short chain $n$-alkanes in the range $C_{14}$–

$C_{22}$, while n-$C_{17}$ is an indicator for algae and photosynthetic bacteria (Aichner et al., 2010; Ficken et al., 1998; Grimalt and Albaigés, 1987; Han and Calvin, 1969). Due to the large range of possible chain lengths present within sediments, ratios of $n$-alkanes are often used to determine the vegetation distribution. Commonly used ratios are the average chain length (ACL) (Poynter and Eglinton, 1990), the carbon preference index (CPI) (Bray and Evans, 1961), the submerged versus emergent aquatic plants predominance ratio ($P_{aq}$) (Ficken et al., 2000), and the vegetation change ratio (Norm31) (Carr et al., 2014).

Nevertheless, it is still unclear to what extent variations in leaf wax composition within paleoenvironmental archives can be explained in terms of changes in the relative proportions of different plant species and/or the reaction of a plant community to environmental conditions (Carr et al., 2014; Diefendorf and Freimuth, 2017).



Determination of human presence in lake catchments often relies on anthropological evidence, but advances in proxy development during the past two decades now allow quantification of the presence of humans or pastoralism through steroid fecal biomarker concentrations (Bull et al., 2002). FeSts, such as stanols and bile acids, in lake sediments reflect grazing in a lacustrine catchment (D'Anjou et al., 2012). Specific FeSts such as 5β-stanols are organic compounds produced by the

microbially mediated alteration of cholesterol in the intestinal tracts of most mammals, making them ideal fecal biomarkers (Dubois and Jacob, 2016). Coprostanol and stigmastanol derive from hydrogenation of cholesterol and stigmasterol by bacteria present in the intestines of humans or animals and can indicate human presence and animal husbandry, respectively (Daughton, 2012; Vane et al., 2010). These molecules are also used as chemical indicators of fecal pollution of lakes, rivers, and drinking water (Daughton, 2012; Vane et al., 2010; Wu et al., 2009). In addition, FeSts can originate from vegetation, e.g. β-sitosterol

is synthesised by higher vascular plants (Nishimura and Koyama, 1977; Vane et al., 2010) and its derivative β-sitostanol is generated from a reduction reaction in sediments (Martins et al., 2007).

In this study, we reconstructed fire activity and vegetation changes using a multi-proxy analytical approach applied to lacustrine sediment samples from the south-eastern Tibetan Plateau. This is the first study to combine MAs, PAHs, *n*-alkanes

and FeSts analyses into a single analytical method highlighting the interaction between fire, climate and vegetation during the Holocene. This combination of proxies, when synthesized with regional climate records, helps determine the changing role of local and regional fire activity throughout the Holocene.

## 2 Study Area

The Qinghai-Tibetan Plateau is a vast plateau in central Asia with an average elevation of approximately 4500 m above sea

level (a.s.l.). The TP stretches nearly 1000 km north to south and 2500 km east to west, covering an area of 2 x $10^6$ km² (Dong et al., 2010). In addition to this wide geographic range, the TP also encompasses altitudes ranging from 1500 to > 8000 m a.s.l., resulting in a broad diversity of landscapes with considerable biodiversity. In general, however, vegetation across much of the TP is dominated by meadow, steppe, and shrubs with increasing species richness with increasing altitude (Shimono et al., 2010). The TP is a pivotal research area due to its sensitivity to century-scale or short-term climatic changes and its

influence on global climate (Liu et al., 1998). On the other hand, its remote character limits access to possible paleoclimate archives, resulting in relatively few investigations into species diversity and plant communities (Wang et al., 2006).

The TP's climate is regulated by the critical and sensitive junction of four climatic systems (image S1 in the Supplementary Information): the Westerlies; the East Asian Monsoon; the Siberian cold polar airflow (or Winter Monsoon); and the Indian

Monsoon (Dong et al., 2010). Westerly winds and the Indian Summer Monsoon (ISM) are considered to be the major wind patterns by which atmospheric particulate derived from biomass burning reaches the plateau (Yao et al., 2013). Millennial-scale changes in insolation over the TP affect monsoon variability and the associated moisture reaching the TP. Generally,



during periods of increased insolation, the monsoon extended farther north on the TP, resulting in more vegetation. During decreased insolation, colder, drier conditions dominate the TP and regions influenced by the ISM are restricted to more southerly portions of the Plateau, including the study area. During the late Pleistocene (~ 16 cal ky BP), a cold and dry climate resulted in desert-steppe vegetation across much of the TP (Tang et al., 2000). Global paleoclimate studies indicate that this

last glacial period concluded with a sudden warming event at ~ 15 cal ky BP (Severinghaus and Brook, 1999). The subsequent transition to the Holocene was characterized by increasing temperature and precipitation that enhanced permafrost and snow melting and facilitated tree growth in the TP after 12 cal ky BP (Saini et al., 2017; Tang et al., 2000). This period was depicted by frequent oscillations between warm and cold phases, in Tibet as well as in other parts of the world (Liping Zhu et al., 2008; Liu et al., 2008, 2009). For example, Tang et al. (2000) suggest that the evolution of ISM has considerably fluctuated

throughout the Holocene. Lake Ximencuo (eastern Tibet) sediments record cold events occurring between 10.3 – 10.0, 7.9 – 7.4, 5.9 – 5.5, 4.2 – 2.8, 1.7 – 1.3 and 0.6 – 0.1 cal ky BP, where the cold event at 4.2 cal ky BP had the most substantial impact (Miao et al., 2015; Mischke and Zhang, 2010). Even with these oscillations, the general temperature trends affecting the TP include warm and humid climate in the early to mid-Holocene, as registered in sediments and dust deposits (Liu et al., 2008), and then a cooling trend during the mid-Holocene. The high temperatures during the early Holocene accelerated evaporation

and caused many Tibetan lakes to evolve from open freshwater systems to saline lakes (Dong et al., 2010), despite increased monsoonal precipitation (Bird et al., 2014). During the mid to late Holocene, warm-wet conditions shifted towards a cooler and drier climate, due to weaker solar insolation, and after 5 cal ky BP temperature and precipitation decreased linearly (Bird et al., 2014; Dong et al., 2010; Liu and Feng, 2012; Tang et al., 2000). More recently, human activities and related climate change have significantly altered the regional hydrology and ecosystem functions of the plateau, with degeneration of

vegetation and grassland that led to desertification and frequent dust storms (Wang et al., 2008).

Paru Co (0.1 km$^2$) is located in the Nyainqentanglha Mountains (29°47'45.6"N, 92°21'07.2"E; 4845 m a.s.l.; Figs 1a and 1b) and is dammed by moraines from past glaciations in its watershed. The biome surrounding Paru Co is temperate subalpine steppe, where the lake is located near the border of alpine coniferous forest and tropical and seasonal rainforests (Li et al.,

2016). The lake's watershed is 2.97 km$^2$ and consists of a sloping glacial valley measuring 0.5 to 2.0 km with lateral mountain crests higher than 5000 m a.s.l. A central ephemeral stream channel and a second incised channel drain the lake's watershed and feed Paru Co with runoff. Outflow from the lake drains via a small stream channel located approximately 430 m west of the primary outlet (Bird et al., 2014). The Tropical Rainfall Measuring Mission data (TRMM) from 1998 to 2007 show that approximately 92 % of mean annual precipitation (MAP; 1118 mm y$^{-1}$) at Paru Co occurs between April to September during

the ISM season (Fig. 1c, Bird et al., 2014). Previous paleoclimate work at Paru Co (Bird et al., 2014) indicates the occurrence of intense ISM rainfall between 10.1 and 5.2 cal ky BP, when five-century-long high lake levels were recorded. The ISM weakened after ~ 5.2 cal ky BP, with the exception of a pluvial event centred at 0.9 cal ky BP. Nir'pa Co, a small lake located near Paru Co, suggests a wet period between 3.3 and 2.4 cal ky and drier conditions from 2.4 to 1.3 cal ky, due to lower silt and lithic content, coincident with elevated sand and clay content and lower lake levels (Bird et al., 2017).



## 3 Methods

### 3.1 Coring and chronology

Paru Co core B11 was collected in 2011 and extends from 0 to 435 cm. Seven radiocarbon ages determined by accelerator mass spectrometry (AMS $^{14}$C) were measured on seven carbonized grass fragments and one oogonia sample extracted from the surrounding sediments (Bird et al., 2014). The sedimentation rate is approximately 0.35 mm y$^{-1}$ between 10.7 cal ky BP and the present. Between 10.9 to 10.7 cal ky BP sedimentation rates are approximately ten times higher (3.3 mm y$^{-1}$). The final age-depth model (Fig. 1d) was constructed using a linear regression between 434.9 and 364.1 cm and by fitting a 3$^{rd}$ order polynomial to the AMS $^{14}$C, $^{137}$Cs (-0.013 cal ky) and sediment-water interface (-0.061 cal ky) ages between 364.1 and 0.0 cm. The associated model error is between 15 and 90 years (see Bird et al. (2014) for further details).

### 3.2 Sample treatments

Sub-samples (n = 72) were selected from the core every 5 cm, spanning from 10.9 to 1.3 cal ky BP with a time resolution of about 130 years on average. Unfortunately, the uppermost samples covering the more recent period (1.3 – 0 cal ky BP) have not been processed for this study due to lack of sufficient sample amounts. The samples were sealed in plastic bags and stored at -20 °C, weighed, freeze-dried, ground and reweighed in order to assure ~ 1 g of dry material, allowing the possibility of determining MAs, PAHs, $n$-alkanes and FeSts from the same sample. All samples were ground using a Mixer Mill MM 400 (Retsch GmbH, Germany) ball miller.

The 72 Paru Co samples were extracted with a 9:1 v/v mixture of ultra-grade (Romil Ltd., Cambridge, UK) dichloromethane and methanol (DCM:MeOH) with Thermo Scientific Dionex ASE 350 (Accelerated Solvent Extractor system), in order to extract both the polar and non-polar compounds. For each extraction, we used 22 mL steel cells containing a 27 mm ø cellulose filter, diatomaceous earth, the sample, ~ 2 g of Na$_2$SO$_4$ (to remove residual moisture) and ~ 2 g of activated copper (to remove sulphur that can interfere with PAHs analysis). We added the following internal standard solutions into each cell: 100 µL of $^{13}$C labelled levoglucosan at 1 ng µL$^{-1}$ of concentration, 100 µL of hexatriacontane at 40 ng µL$^{-1}$, 100 µL of a mixture of $^{13}$C labelled PAHs (acenaphthylene, phenanthrene and benzo[a]pyrene) at 1 ng µL$^{-1}$, 100 µL of cholesterol-3,4-$^{13}$C$_2$ at 1 ng µL$^{-1}$. The extractions were performed with three static cycles at 100 °C and 1500 psi. A procedural blank was created and extracted for every batch of 12 samples, where we filled the steel cell with all of the same reagents, but without a sample. Each sample was then purified with three steps to obtain a PAHs/$n$-alkanes fraction, a FeSts fraction and a MAs fraction. We combined and modified published clean-up methodologies in order to obtain the necessary fractions (Battistel et al., 2015; Douglas et al., 2012; Kirchgeorg et al., 2014; Martino, 2016). Our resulting method uses 12 mL Solid-Phase Extraction cartridges (SPE DSC-Si 10 Tube; 12 mL; 52657 Supelco, Sigma-Aldrich) packed with 2 g of silica gel (particle size 50 µm) and installed on Visiprep™ (SPE Vacuum Manifold standard, Sigma-Aldrich) to accelerate purification. We conditioned each cartridge with 30 mL of DCM and 30 mL of Hexane (Hex). The first non-polar fraction (F1), containing PAHs and $n$-alkanes, was eluted



using 40 mL of a Hex:DCM 9:1 v/v mixture. Then, the second polar fraction (F2), containing FeSts, was separated with 70 mL of DCM. This fraction was derivatized, according to Battistel et al. (2015), at 70 °C for 1 h with 100 μL of BSTFA + 1% TMCS (N,O-bis(trimethylsilyl)trifluoroacetamide with 1 % trimethylchlorosilane, Sigma-Aldrich) to increase compound volatility and detectability during gas chromatography – mass spectrometry (GC-MS) analysis. Finally, the third polar fraction

(F3) containing MAs was eluted with 20 mL of MeOH. F1 and F2 were evaporated under a stream of pure $N_2$ using a TurboVap II® system (Caliper Life Science, Hopkinton, MA, USA) in order to reduce the volume to 150 μL. F3 was dried, dissolved in 0.5 mL of ultra-pure water and sonicated to avoid any adsorption of MAs to walls of glass evaporation tubes. Finally, the samples were centrifuged (5 min, 14000 rpm) and transferred using decontaminated Pasteur pipettes to the measurement vials.

### 3.3 Sample analysis

MAs were detected using methods published in Kirchgeorg et al. (2014) with Ion Chromatography (IC Dionex ICS 5000, Thermo Scientific, Waltham, USA) coupled with a single quadrupole Mass Spectrometer (MSQ Plus™, Thermo Scientific) equipped with CarboPac MA1™ column (Thermo Scientific, 2 x 250 mm) and an AminoTrap column (2 x 50 mm), resulting in a good separation of the isomers levoglucosan, mannosan and galactosan. The injection volume was 50 μL. A solution of MeOH/$NH_4OH$ was added post-column (0.025 mL min$^{-1}$) to improve ionization of the aqueous eluent before entering the

electrospray ionisation (ESI) in negative mode. The analytes were quantified according to specific mass to charge ratios and with calibration curves and response factors containing unlabelled molecules of L, M, G, as well as an internal standard molecule ([13]C labelled levoglucosan).

The seventeen priority PAHs (according to US ATSDR, 1995) plus retene, $n$-alkanes (from $C_{10}$ to $C_{35}$) and FeSts (coprostanol,

epi-coprostanol, cholesterol, 5α-cholestanol, sitosterol, sitostanol) were analysed with Gas Chromatography (6890-N GC system) coupled to a single quadrupole Mass Spectrometer (MS 5975, Agilent Technologies, Santa Clara, CA, USA) (Argiriadis et al., 2014; Battistel et al., 2015; Gregoris et al., 2014; Martino, 2016; Piazza et al., 2013). Each analysis used the same capillary column (HP5-MS (5%-phenyl)-methylpolysiloxane, Agilent Technologies, Santa Clara, CA, USA). The conditions were an injection volume of 2 μL (split valve open after 1.5 min) and He as a carrier gas (1 mL min$^{-1}$). The MS was

equipped with an electronic impact (EI) source used in positive mode. The analytes were quantified in single ion monitoring mode (Table 1). We created response factors containing all of the target compounds as well as internal standards molecules ([13]C labelled acenaphthylene, phenanthrene and benzo[a]pyrene; hexatriacontane; cholesterol-3,4-[13]$C_2$). We ran a response factor after every seven samples in order to monitor possible instrumental drift, as well as running a full calibration curve of external PAHs standards before each set of analyses.

Target molecules with respective analysed ion and method detection limit (MDL) are listed in Table 1. Further method details and quality assurance can be found in previously published works (Argiriadis et al., 2014; Battistel et al., 2015; Gregoris et al., 2014; Kirchgeorg et al., 2014; Martino, 2016; Piazza et al., 2013). Chromatographic peak identification and calculations



were performed using the Chromeleon™6.8 Chromatography Data System Software (Thermo Scientific, Waltham, USA) and Agilent G1701DA GC/MSD ChemStation (Agilent Technologies, Santa Clara, CA, USA).

**3.4 Data elaboration**

Data elaboration and statistics were performed with Microsoft Excel, R and OriginPro 8. All the concentration values obtained from IC and GC-MS analysis were converted in ng g$^{-1}$, using the dry weight of each sample, and then transformed into fluxes in order to correct the data for the influence of time and sedimentation. Fluxes (ng cm$^{-2}$ y$^{-1}$) were calculated by multiplying sedimentation rate (cm y$^{-1}$), wet density (g cm$^{-3}$) and concentration (ng g$^{-1}$) of the respective analyte (Menounos, 1997). 2-tailed Pearson's correlations were calculated in R with a 95% confidence interval (chart S3 in the Supplementary Information) with statistically significative results when p-value < 0.05.

$N$-alkanes ratios useful for work include: the average chain length (ACL), representing the composite of longer and shorter $n$-alkanes (Poynter and Eglinton, 1990) and encompassing the chain length range of 21 to 33; the P-aqueous ratio (P$_{aq}$), that can help differentiate between submerged plants that tend to have medium-chain-length $n$-alkanes and terrestrial plants that tend to have longer chain lengths (Ficken et al., 2000); and the Norm31 ratio (Carr et al., 2014), that specifically examines changes in the distribution of the longest-chain n-alkanes to identify general changes in vegetation types. These ratios were calculated according to the following equations:

(1) $\quad ACL_{21-33} = \frac{\sum(n_{21-33})(C_{21-33})}{\sum(C_{21-33})}$

(2) $\quad P_{aq} = (C_{23} + C_{25})/(C_{23} + C_{25} + C_{29} + C_{31})$

(3) $\quad Norm31 = C_{31}/(C_{29} + C_{31})$

where n$_{21-33}$ indicates the number of carbons in the $n$-alkanes chains and C$_n$ represents the concentration of the respective $n$-alkane.

Charcoal is the most widely used fire proxy, and we therefore compiled regional charcoal data from the Global Charcoal Database (GCD) as regional syntheses of past biomass burning events in the TP. The paleofire R library (Blarquez et al., 2014) entails version 3 of the database (Marlon et al., 2016). The range of metrics used to quantify charcoal (e.g., influx, concentration, charcoal/pollen ratios, gravimetric, image analysis, etc.) results in individual data values that vary over 13 orders of magnitude among and within sites, requiring standardizing data between sites (Power et al., 2010). The standardization protocol used for obtaining the charcoal index is described by Marlon et al. (2008) and Power et al. (2010). After selecting latitude and longitude ranges of regional charcoal data, this compilation provides an independent fire history that can be visually compared with the MAs fluxes in the Paru Co core.



## 4 Results

### 4.1 Paleofire indicators

Both groups of fire molecular markers in the Paru Co core – MAs and PAHs – demonstrate similar trends in biomass burning activity (Fig. 2a). The MAs record from Paru Co spans from 10.9 to 1.3 cal ky BP, with a major MAs peak in the early

Holocene, with a sharp decrease from 8.5 cal ky BP and then a long decreasing trend to 1.3 cal ky BP. This major peak in MAs values for the samples in the period 10.9-10.8 cal ky BP, coincides with the highest sedimentation rates in the core, suggesting that MAs may have been better preserved due to the high quantity of sediment transported into the lake. In order to better visualize MAs trends, the deepest samples were omitted in Fig. 2b, making it possible to identify the fire peaks at approximately 10, 8.6, 5.6 and 2.5 cal ky BP. Over the entire core, MAs fluxes range from 5 to 2500, from 0.7 to 162 and from 0.9 to 531 ng

$cm^{-2}$ $y^{-1}$ for levoglucosan, mannosan and galactosan respectively. The Paru Co MAs results reflect the general observation in the literature that mannosan and galactosan concentrations are almost always less than levoglucosan concentrations, which may be due to the different thermal stability of their precursors, hemicellulose and cellulose, respectively (Kuo et al., 2011; Simoneit, 2002). Although levoglucosan and galactosan may have different precursors, their trends throughout the Paru Co core are generally similar, while peaks in mannosan concentrations slightly differ from the other two isomers.

The fluxes of the total sum of the PAHs congeners mostly vary between 0 and 20 ng $cm^{-2}$ $y^{-1}$and contain peaks up to 110 ng $cm^{-2}$ $y^{-1}$ (Fig. 2a). Within the 18 analysed congeners of PAHs, naphthalene, fluorene, fluoranthene, phenanthrene and benzo[e]pyrene reach the highest concentrations, touching values till 70 ng $cm^{-2}$ $y^{-1}$, primarily between 10.9 and 10.7 cal ky BP. These high peaks of fire activity in the oldest part of the record are similar to the fire history recorded by MAs. The other

PAHs had low fluxes near or below MDL (image S2 in the Supplementary Information). Plotting only the samples above the deepest and oldest samples, allows observing trends in PAHs that are otherwise obscured by the high peaks in the deepest part of the core (Fig. 2b). The PAHs record differs from the MAs fire history after the initial major peak in biomass burning; MAs steadily decline throughout the Holocene while PAHs do not mirror this long-term trend.

MAs and PAHs were then plotted together with biogenic silica (BSi) and total organic matter (TOM) data (Bird et al., 2014) to identify if influences from the monsoon or from the association with organic materials, respectively, could have had a role in their distribution in the sediment core (Fig. 3). Pearson's correlation coefficients (chart S3 in the Supplementary Information) were calculated for the periods 10.7-1.3 cal ky BP between MAs and % of BSi and between MAs and % of TOM, with negative results, i.e. r = -0.54 (p-value $3.46*10^{-6}$) and r = -0.38 (p-value 0.00175) respectively. This anti-correlation can also be visually

observed in Fig. 3 and indicates the non-dependence between MAs quantities and the organic content of the sediments. The TOM only slightly varied displaying a modestly increasing trend across the Holocene. The BSi signal reflects changes in the ISM, with warmer and wetter climate between 11 to 5 cal ky BP followed by cooler and drier conditions (Bird et al., 2014, 2017).



Throughout the Holocene, the obtained data for ∑PAHs and ∑MAs do not correlate (r = -0.06, p-value 0.665). However, ∑PAHs and ∑MAs positively correlate during the early Holocene (10.9 – 8.7 cal ky BP), with r = 0.51 (p-value 0.029), while in the following period (8.7 – 1.3 cal ky BP) a slight, but not significant, negative correlation is observed, with r = -0.26 (p-value 0.08). Considering that the MAs catchment area varies from local to regional scales, the high early Holocene (10.9 – 10.7 cal ky BP) fluxes of MAs and PAHs may be explained with local fire activity, better preservation and higher original concentrations in the sediments. Then, in the period 10.7 – 8.7 both local and regional fire origin may be hypothesised. After 8.7 cal ky BP PAHs may have had a biogenic origin, due to the fact that TOM and PAHs show increasing trends from 8 to 1.3 cal ky BP.

High percentages of 4-6 ring PAHs generally suggest the contribution of local high-temperature combustion (Yang et al., 2016), where such combustion may be the source of benzo[e]pyrene, the congener with the second highest concentration in Paru Co. PAHs can also serve as local indicators of their sources, where the organic carbon content can be compared to the presence of PAHs in soils (Abdel-shafy and Mansour, 2016). Both PAHs and TOM trends in Paru Co slightly increase from 8.3 to 1.3 cal ky BP (Figs. 3c and 3d). However, this fact is not statistically confirmed (r = 0.26, p-value 0.08), and may be due to the noisy trend in PAHs. In general, the low amounts of PAHs in Paru Co may derive from their partial solubility in water, especially for the lower molecular weight PAHs (Abdel-shafy and Mansour, 2016). The relative proportions of PAHs originating from land areas are generally independent of the sedimentation rate (Zakir Hossain et al., 2013). Therefore, the relative composition of the PAHs could record details of changes in the terrigenous environment surrounding Paru Co (Matsubara Pereira, 2017), but further investigation is needed to explore these results.

### 4.2 Vegetation and human indicators

The ratios of both MAs and *n*-alkanes help reconstruct past vegetation. MAs ratios can help determine past vegetation types and/or burning temperatures. Higher combustion temperatures (∼ 300 °C) and longer combustion duration result in higher L/M and L/(M+G) ratios, regardless of plant species (Kuo et al., 2011). The L/M and L/(M+G) ratios in Paru Core range from 0.6 to 100 and 0.5 to 11.1, respectively. The L/M ratios peak between ∼ 6 to 7 cal ky BP; the L/(M+G) values do not peak at the same time, but oscillate throughout the Holocene, with the highest values centred around ∼ 2 cal ky BP (Figs. 4a and 4b). Although MAs ratios cannot precisely point to the type of past burnt vegetation, they can classify general vegetation categories. According to their published ranges (Fabbri et al., 2009), our data suggest that grasses dominated the area for the oldest section of the Paru Co core and that softwood began to grow in the region after ∼ 10.74 cal ky BP. Grasses, softwood and hardwood may have oscillated until 8.6 cal ky BP. Hardwood generally dominated the vegetation between 8.6 to 7.7 cal ky BP, followed by primarily grasslands until the present. Even though MAs ratios can generally differentiate between grass versus wood burning (Kirchgeorg et al., 2014), specific L/M and/or L/(M+G) ratios do not directly correspond to individual fuel types (Matsubara Pereira, 2017) due to the problem of overlapping values and similar burning conditions that influence the ratios.



Past vegetation changes can also be derived by variations in *n*-alkane ratios, as *n*-alkanes record the organic input into the lake. Rapid fluctuations in $ACL_{21-33}$ values at 10.9 – 10 cal ky BP may reflect quick transitions between terrestrial and aquatic vegetation, while decreased $ACL_{21-33}$ values between 10-5.5 cal ky BP may result from the prevalence of submerged aquatic

plants (Fig. 5b), and then prevalence of $C_{27}$-associated terrestrial vegetation from 5 to 1.3 cal ky BP. The $P_{aq}$ ratio values closer to 1 indicate a greater percentage of submerged plants, and when the value is closer to 0, these numbers pertain to a greater percentage of terrestrial vegetation. The Paru Co $P_{aq}$ ratio (Fig. 5c) quickly oscillates in the oldest section of the core, suggesting rapid changes between terrestrial and aqueous vegetation as the dominant source of *n*-alkanes to the lake, as also supported by $ACL_{21-33}$ data.

The Paru Co Norm31 ratios demonstrate alternating vegetation types throughout the Holocene (Fig. 5d). High sitostanol fluxes occur at similar core depths as high Norm31 ratios (Fig. 5e), suggesting that the vegetation may be one of the possible sources of sitostanol in the core. The only FeSts that were above the MDL in the Paru Co samples are sitostanol and sitosterol, where these FeSts highly correlate with each other (r = 0.94, p-value $3.79*10^{-8}$) due to the fact that these molecules can be produced

by plants and reduction reactions in sediments. All other FeSts were either not present and were below MDL in Paru Co samples, suggesting the virtual absence of local humans and grazing animals in the lake catchment throughout the examined time period.

## 4.3 GCD results

The GCD version 3 allows extracting and compiling individual charcoal records into user-defined syntheses using the paleofire

R package. Statistical methods to create these compilations are described in detail in Blarquez et al. (2014). The GCD does not include charcoal records located in the TP, but does include records from elsewhere in China and Asia. Here, we selected all available charcoal sites with a radius of ~ 1000 km from Paru Co, and that encompass the time period between 0 and 12 cal ky BP, resulting in a total of 43 sites (Fig. 6a). Due to the wide geographic distribution of these sites, these charcoal records are located in a variety of elevations and ecosystems. The catchment area of an individual charcoal record is only a few km,

but this synthesis results in charcoal records across 1000s of km. In addition to only traveling a few km, macroscopic charcoal particles (> 100 μm ) usually result from burning at temperatures between 250 and 550 °C , while levoglucosan is produced at temperatures centred around 250 °C and can travel up to thousands of kilometres from its source (Schüpbach et al., 2015; Zennaro et al., 2014). When comparing the charcoal and MAs records the differences in catchments may explain much of the dissimilarities in trends over a millennial scale (Fig. 6). Dating uncertainties among the different records can be carried over

into the composite charcoal index and can be a source of misinformation. However, some similarities exist in the short-term variability, as highlighted with coloured bars and arrows in Figs. 6b and 6c. The comparison at this level is merely visual, due to the fact that MAs values are presented in fluxes while charcoal is displayed as a standardised index.



## 5 Discussion

### 5.1 Paleofire activity

The TP is ringed by high mountains that create natural barriers that block the transport of smoke aerosols to the TP from the south, west, and northwest (You et al., 2016a). However, the Indian Summer Monsoon may help transport both mineral and organic aerosols over the mountain ridges and into the TP during the summer monsoon months when winds rush from the south across the Himalayas. The ISM is the main source of precipitation across much of the southern TP, where this rainfall provides moisture for plant growth. The strength of the ISM over millennial timescales is driven by solar radiation, where increased insolation results in the ISM moisture moving northward across the TP. Climatic records from areas surrounding the TP demonstrate that the Pleistocene-Holocene transition was characterised by increasing temperatures until approximately 8.2 cal ky BP, when sudden cooling occurred (Mischke et al., 2016). The ISM was more intense than current levels between ~ 10 – 6 cal ky BP due to increased insolation, and reached a maximum in the south-eastern TP at 8 cal ky BP (Tang et al., 2000). The mid-Holocene had higher average summer sea surface temperatures (SST) and a stronger summer monsoon than during the present, resulting in warm and wet climate (Wei et al., 2007; Zhao et al., 2011). This timing is consistent with paleo-monsoon records from southern China and with the idea that the interplays between summer insolation and other large-scale boundary conditions, including SST and sea-level change, control regional climate (Zhao et al., 2011). A drying trend during the past 6 cal ky is documented in many records from the northern subtropics and tropics (Liu and Feng, 2012). The cooling trend after the Holocene Climatic Optimum (6.5-4.7 cal ky BP) correlates with decreasing solar insolation (Zhao et al., 2011) and caused a progressive southward shift of the northern hemisphere summer position of the Inter Tropical Convergence Zone, resulting in a decreasing strength of the Asian monsoon systems and in a drier climate across much of the TP. Decreased solar insolation resulted in a dramatic drying at ~ 4.2 cal ky BP, directly or indirectly leading to the observed collapses of many Chinese Neolithic cultures (Liu and Feng, 2012; Wang et al., 2005). During the past 750 years, precipitation changes in the Altai controlled fire-regime and vegetation shifts, and the high sensitivity of ecosystems to occasional decadal-scale drought events may, in the future, trigger unprecedented environmental reorganization under global-warming conditions (Eichler et al., 2011).

The Paru Co fire record demonstrates similar fire histories in the early Holocene for all three proxy types. During the time interval ~ 10.9 – 9.5 cal ky BP, MAs, PAHs and the regional charcoal composite all have elevated fluxes, suggesting increased regional fire activity (Figs. 2 and 6). These observed high levels of fire recorded in Paru Co are consistent with weaker ISM in the period 10.9-10.7 cal ky BP (Bird et al., 2014) and with the idea that dry conditions in Asia before 10 cal ky BP supported biomass burning, higher in the early Holocene than in the late Holocene (Marlon et al., 2013). However, when the monsoon reaches its peak at ~ 8 cal ky BP, the Paru Co fire records and regional charcoal composite substantially differ from one another (Figs. 6b and 6c). The MAs record demonstrates high fire activity between 8 to 9 cal ky BP, while the PAHs and charcoal records demonstrate decreased fire activity during this time period. These differences may be due to factors such as different



catchment sizes (as discussed in Section 4.3) as MAs are a regional record, while charcoal provides specific information for a local point, where an individual record may influence the results of a compilation. In addition, many of these charcoal records are located far from Paru Co. However, the PAHs record is from the same Paru Co core as the MAs. The difference between these two records may be influenced by the burning temperatures that produce the different products, where MAs may reflect

low temperature fires around ~ 250 °C (Zennaro et al., 2015 and references therein). Although the ISM reached its maximum at ~ 8 cal ky BP (Tang et al., 2000) resulting in relatively wet conditions with the potential to provide more vegetation growth and hence biomass source for fire (Bird et al., 2014; Marlon et al., 2013), within this generally humid period, the abrupt climate event at ~ 8.2 cal ky BP brought generally cold and dry conditions to much of the northern Hemisphere (Alley et al., 1997; Alley and Ágústsdóttir, 2005; Barber et al., 1999). In the Tibetan area, a dry interval at Shumxi Co, associated with the 8.2 cal

ky event, is indicated by pollen and diatom records (Van Campo and Gasse, 1993). A peak in aeolian silt in north-western China (Lop Nur, Xinjiang) is also associated with the 8.2 cal ky event (Liu et al., 2003), and correlates with a cold interval in pollen diagrams from Qinghai Lake (Koko-nur) (Alley and Ágústsdóttir, 2005; Liu et al., 2002). The Paru Co fire peaks between 8 – 8.5 cal ky BP may therefore be associated with this cold, dry climate event following a period of enhanced vegetation growth.

The centennial-scale variability of monsoon precipitations is also characterized by events that correlate with changes in oceanic and atmospheric circulation (Bond et al., 2001; Wang et al., 2005). Recent studies have shown that the climatic change at 5.5 cal ky BP (Bond event 4) was one of the most prominent Holocene climatic events that affected much of the world in the Holocene (Liu and Feng, 2012; Wei et al., 2007). Regional fire history increases during this time period, with elevated MAs

fluxes in Paru Co, increased regional charcoal, as well as high charcoal concentrations in the south-central TP Nam Co core (Herrmann et al., 2010).

In general, the Paru Co fire history shows a decreasing trend from 8 cal ky BP to the present, which is consistent with the diminishment of the ISM. This decreasing fire pattern observed in Paru Co may be associated with a regional cooling trend

reconstructed from Lake Zigetang with the pollen ratio *Artemisia*/Cyperaceae, a semi-quantitative measure for summer temperature, indicating a general cooling trend throughout the Holocene (Herzschuh et al., 2006). As already mentioned in Section 4.1, the noisy PAHs signal in Paru Co after 8 cal ky BP may not be not fire related but instead may be associated to the biogenic/diagenetic transformation of natural organic matter in the lake (Saber et al., 2006; Stogiannidis and Laane, 2015). Figure 3 shows increasing trends both in PAHs and TOM from 8 to 1.3 cal ky BP, evidencing the possible association of these

two variables.

Regional dust records can provide information regarding past wind speed and direction. However, the transport and source of mineral dust versus the transport and source of organic fire markers may differ, and we would like to highlight that increased dust does not imply increased fire frequency. Dust layers in Genggahai Lake demonstrate weak aeolian activity between 10.3



to 6.3 cal ky BP (Qiang et al., 2014), which may be a response to increased vegetation cover due to the strengthened Asian summer monsoon. In contrast, in central Asia (Lake Zhuyeze, Mischke et al., 2016) the 8.2 cal ky event increased the mobility of aeolian sands which gradually caused the degradation of vegetation because of burial and led to massive and widespread aeolian sand transportation, until ~ 7.5 cal ky BP when the vegetation recovered. Dust layers occurred episodically when the

summer monsoon weakened in the periods 6 – 5.5, 4.6 – 4, 1.8 – 1.4 and 0.2 cal ky BP and these abrupt events of sand mobility were associated with enhanced wind strength, probably in response to cooling events at high latitudes (Qiang et al., 2014). This increased aridity, coupled with increased winds, may have influenced the transport of MAs to the analysed lake, as reflected in the fire peak at 5.6 cal ky BP at Paru Co.

### 5.2 Past vegetation reconstruction

Isotopic and pollen information from surrounding lakes support the climatic variation from a cold-dry early Holocene to a warm-humid mid to late Holocene and also ascribe these climate changes to the ISM (Kramer et al., 2010a, 2010b; Ma et al., 2014; Zhu et al., 2010). Pollen assemblages from two transects of lakes (east-west and north-south) across the TP indicate sparse vegetation with low pollen concentrations characterized by *Artemisia*/Cyperaceae alpine steppe (Li et al., 2016). Lake Naleng, also located on the south-eastern TP, records changes that are similar to Paru Co paleoreconstructions (Kramer et al.,

2010a). From 10.7 to 4.4 cal ky BP open *Abies–Betula* forests reflect intense summer monsoon and an upward treeline shift. Temperature range reconstructions demonstrate climate 2 – 3 °C warmer than present and treeline position 400 – 600 m higher than today. However, within this warm period, the climate had a sudden, intense change between 8.1 and 7.2 cal ky BP with temperatures 1 – 2 °C below early and mid-Holocene levels and forests retreating downslope (Kramer et al., 2010a). Multiple pollen studies confirm the severe early Holocene cold events at 8.7 – 8.3 and 7.4 cal ky BP (Miao et al., 2015; Mischke and

Zhang, 2010). During the mid-Holocene (7.3 – 4.4 cal ky BP), dense temperate steppe vegetation dominated the TP (Li et al., 2016; Zhao et al., 2011). Tree pollen (primarily *Picea*) peaks during the mid-Holocene at 6.5 cal ky BP, and then decreases until 2 cal ky BP (Zhao et al., 2011). During the same time period, Cyperaceae becomes the dominant regional steppe vegetation, and altitudinal vegetation belts shifted downslope in response to reduced temperatures (Li et al., 2016). These alpine steppes contain desert vegetation elements (a composite of Cyperaceae, Poaceae, Chenopodiaceae, and characteristic

high-alpine herb families) between 4.4 – 0 cal ky BP (Herzschuh et al., 2006; Tang et al., 2000). Lake records from Nam Co and Taro Co, located in the same vegetation zone as Paru Co, suggest a weakening in the ISM and the increased influence of the westerlies from 5.6 to 0.9 cal ky BP (Bird et al., 2014; Li et al., 2011; Ma et al., 2014). This synthesis on changes in Holocene vegetation suggests that variations of monsoonal precipitation and insolation-driven temperature are the predominant driving forces for changes in alpine vegetation in the central TP (Li et al., 2016).

A regional synthesis of pollen records along a south-north transect indicates that climate and vegetation reliance on the monsoon through time is prevalent across the eastern TP (Zhao et al., 2011) suggesting that Paru Co paleoreconstructions may also be considered within this framework. In general, vegetation density and productivity increased during the early to mid-



Holocene, as suggested by relatively high pollen concentrations in the Dunde ice cap (northern TP) during ca. 10 – 4.8 cal ky BP, with a limited abrupt to the previously-mentioned dry conditions during 8.0 – 7.7 cal ky BP (Liu et al., 1998). Pollen data are consistent with the ice core oxygen isotope record showing a gradual $^{18}$O enrichment throughout the Holocene with a warm period centred at 8 – 6 cal ky BP. Finally, the intervals at 2.7 – 2.2, 1.5 – 0.8 and 0.6 – 0 cal ky BP were comparatively humid

periods with higher vegetation density and productivity, with characteristic pollen taxa of alpine meadow (Cyperaceae, *Polygonum*) (Liu et al., 1998). These studies suggest that increased biomass availability provided fuel for fires and was probably the pivotal driver for the kindling of intense fire activity periods in Paru Co.

A pollen record from the nearby Hidden Lake (Figs. 1 and 4), demonstrating Holocene vegetation fluctuations in the area

(Tang et al., 2000), contains similarities with the Paru Co L/M trend in vegetation changes (Figs. 4b and 4c). These records are consistent with the literature and demonstrate that ~ 8 cal ky BP meadows began to be replaced with softwood and then between 5.3 to 3 cal ky BP, these conifers began to be substituted by steppe vegetation (Tang et al., 2000). Data-model comparisons also help depict TP Holocene climatic trends and insert Paru Co vegetation reconstruction in the Tibetan context. Model reconstructions primarily identify decreasing summer monsoon precipitation and changes in warm season temperature

as the mechanisms responsible for the vegetation shift (Dallmeyer et al., 2011). The average forest fraction on the TP shrank by almost one-third from the mid-Holocene (41.4%) to the present (28.3 %). Shrubs quadrupled in their mid-Holocene percentage to present-day (12.3 %), replacing much of this forest. The grass fraction also increased from 38.1% during the mid-Holocene to the current percentage of 42.3% (Dallmeyer et al., 2011). This forest decline and replacement by shrubs from 6 cal ky to present is prevalent across much of the south-eastern TP (Lu et al., 2011).

Paru Co data demonstrate the correlation between $ACL_{21–33}$ and $P_{aq}$, where prevalence of submerged aquatic plants is associated to lower ACL and higher $P_{aq}$ values. Locally, after 5.2 cal ky BP lake levels decreased probably causing opposite fluctuations in both ACL and $P_{aq}$, suggesting diminished ISM rainfall, reduced clastic deposition and lowered lake levels, which led to an invasion of the littoral zone on the core site and an increase in sand deposition (Bird et al., 2014). Generally,

according to reconstructed changes in summer radiation (Berger and Loutre, 1991), the insolation is suspected to be the major driver for millennial-scale changes in vegetation around Paru Co (Figs. 5a, 5b, 5c). Residues from leaf waxes also help provide additional information on the links between climate, vegetation and fires in the Paru Co region. Modern leaf *n*-alkanes from plants in Qingjiang (Hubei province, China) demonstrate remarkable seasonal variation in their CPI and ACL values (Cui et al., 2008). During warmer months, the CPI values of all plant species decrease gradually due to the fading process of the leaves.

The ACL values are greater in mid-summer than in May and November, suggesting that temperature influences these values. Compared to the fresh leaves, the defoliated leaves have an elevated abundance of *n*-alkanes, possibly due to degradation by microorganisms and associated biotransformation (Cui et al., 2008). We can therefore infer that biotransformation of fresh leaves occurs inside the sediments, increasing the quantities of these biomarkers, which were then detected in high abundances in the Paru Co samples. Moreover, because of the fact that the climate of the Nyainqentanglha Mountains is dominated by the



ISM and deposition of leaves only occurs at Paru Co during the boreal summer when the lake is ice-free, the interpretation of sedimentological, and perhaps also *n*-alkanes, variability reflects summer climatic changes (Bird et al., 2014). As prolonged charring reduces the average chain length of *n*-alkanes by up to four carbons and creates a balanced odd/even distribution of carbons in the leaf waxes (Knicker et al., 2013), seems that *n*-alkanes could help determine fire history. However, significant correlation between *n*-alkanes ratios and MAs ratios is only observed for Norm31 and L/(M+G), with r = 0.36 (p-value 0.009) insinuating a starting point for a more detailed study of the potentiality of MAs ratio.

The decline in forest vegetation and the rise in steppe vegetation seems to coincide with an increased human presence on the TP. Grazing indicators (increases in *Rumex*, *Sanguisorba* and Apiaceae pollens), imply a human influence on the environment since approximately 3.4 cal ky BP near Lake Naleng (Kramer et al., 2010a), in the south-eastern TP, the general region of Paru Co. Humans slashed and burned the forests near Lhasa to open lands through fire (Miehe et al., 2006). Other studies also suggest links between fire activity and forest clearance in the southern and south-eastern TP during the late Holocene (Kaiser et al., 2009a, 2009b). Although evidence exists that humans altered TP vegetation through burning in the late Holocene, the extent of human activity on vegetation change across the TP is still unknown. The absence of anthropogenic FeSts in Paru Co sediments indicates that human and associated pastoralism were not present in the local area. In Paru Co, the only FeSts above the MDL were sitosterol and sitostanol. Sitosterol can derive from higher vegetation, but its lower amounts respect to the sitostanol can indicate the microbial reduction of sitosterol into sitostanol in the stomach of ruminant animals (Vane et al., 2010), as well as its hydrogenation in sediments (Martins et al., 2007). The fact that sitosterol and sitostanol were the only FeSts detected in Paru Co suggests the absence of ruminant animals that would also deposit other FeSts, and we consider vegetation and reduction reactions in sediments as the main source of Paru Co FeSts. Due to the absence of other human/animal indicators we are inclined to describe the variations found in fire regimes and vegetation as primarily climate-driven signals.

The differences in trends between the *n*-alkanes vegetation data, pollen records and MAs ratios may be due to their differing provenance. MAs, in general, record regional fire, except when associated to PAHs, where local biomass burning can be detected. Conversely, *n*-alkanes are local indicators and can also originate from living plants. The *n*-alkanes may differ from pollen as angiosperms produce more *n*-alkanes than do gymnosperms. This discrepancy is the probable reason why *n*-alkanes demonstrate a different grass/wood prevalence than that recorded by MAs ratios and pollens (Figs. 4 and 5). MAs and pollens are able to be transported hundreds to thousands of kilometres, and so primarily reflect regional environmental modifications. Conversely, *n*-alkanes register that vegetation changes in Paru Co may be related to local-scale lake levels fluctuations and to global-scale solar radiation (Fig. 5). Moreover, sitostanol is associated to local degradation of sitosterol derived from higher plants present near the lake. FeSts are markers of local human and animal presence as well, and their absence in Paru Co exclude the anthropogenic influence on the lake environment. Comparing data resulted from this multi-proxy study is helpful to understand past environmental processes happened within the lake and the south-eastern Tibetan Plateau, highlighting how



diverse fire and vegetation markers are needed to obtain both the local and the regional paleoreconstruction information, using the same sediment core.

## 6 Conclusions

This study is the first multi-proxy work on paleofire activity in lacustrine sediments from the TP , and provides a starting point for future investigations in this field of research. The combination of MAs, PAHs, FeSts and *n*-alkanes as fire and vegetation markers, helps reconstruct the biomass burning history of the south-eastern Tibetan Plateau using innovative biomarkers. The results reveal intense climate-induced fire activity in the period 10.9-8 cal ky BP and then a long-term decreasing trend in fire. Vegetation reconstructed from MA and *n*-alkanes ratios was characterised by short-term oscillations and alternating softwood and grasses/steppe vegetation composition, and a long-term pattern due to orbital-induced insolation changes. The apparent absence of human impact indicators as determined by the lack of human FeSts above the MDL, excludes local anthropogenic influence on fire and vegetation changes. Fire and vegetation records in Paru Co are instead primarily driven by climatic factors as follows:

1) Early Holocene: 10.9 – 7.5 cal ky BP. The period 10.9 – 7.5 cal ky BP was characterised by an increasingly warming and drying climate followed by an intensification phase of the ISM until the mid-Holocene. These conditions may have favoured vegetation growth that led to fire activity recorded in Paru Co by both MAs and PAHs, with major peaks of fire activity around 10.9 – 9.5 cal ky BP. Regional charcoal compilations also depict increased biomass burning from 11 to 9.5 cal ky BP. However, the charcoal records cannot be related to the increased fire activity recorded in Paru Co between 9 to 8 cal ky BP. This difference suggests a relatively local fire source for the Paru Co biomass burning, as the closest available charcoal records are still hundreds of kilometres away. The decrease in fire activity may have been affected by the dramatic dry event of ~ 8 cal ky BP. Early Holocene MAs ratios and pollen records indicate meadows as the prevalent regional vegetation in the surroundings of Paru Co, where *n*-alkanes ratios depict vegetation oscillations during much of this time period, as well as submerged/emergent vegetation alternation.

2) Mid-Holocene: 7.5 – 3.8 cal ky BP. Increased forest vegetation cover due to the strengthened ISM is observed in Paru Co, yet the 5.6 cal ky BP fire event is the unique peak recorded in this core in the mid-Holocene. This ~ 5.6 cal ky BP fire peak also occurs in the charcoal composite record too, possibly indicating a regional fire source, as it can be likewise associated with enhanced aeolian activity most likely related to drying and cooling events. Cold and dry conditions reach the maximum at circa 4.2 cal ky BP, coinciding with the expansion of Bronze Age civilizations, whose eventual impact on the landscape is still unknown.

3) Late Holocene: 3.8 – 1.3 cal ky BP. The past few thousand years were characterized by low fire activity in Paru Co, with some weak exceptions around 3 – 2 cal ky. Emergent terrestrial plants, steppe and grasses dominated the vegetation at this site, in correspondence with the weaker summer insolation and the reduced precipitation.



*Acknowledgements*. This work was funded as part of the Early Human Impact project, "Ideas" Specific Programme - European Research Council - Advanced Grant 2010 - Grant Agreement n° 267696, and was part of a PhD program in Environmental Sciences at Università Ca' Foscari. We are grateful to all the colleagues who helped in the lab and all the research team. We thank Aaron Diefendorf and Laura Strickland for their helpful suggestions for improving the work. Any use of trade, firm, or product names is for descriptive purposes only and does not imply endorsement by the U.S. Government.

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



**Figure 1:** (a) Map of the Tibetan Plateau and surrounding territories showing the location of Paru Co (red pin) and of the other lakes mentioned in the text (blue circles): Taro Co (TC), Nam Co (NC), Hidden Lake (HL) and Lake Naleng (LN). (b) Satellite image of Paru Co. (c) Average monthly precipitation at Paru Co based on TRMM data from 1998 to 2007 and average monthly temperatures at Paru Co (4845 m asl) from Lhasa (3650 m asl) weather station data using a lapse rate of $-6.4$ °C km$^{-1}$. (d) Plot of the age/depth model for Paru Co according to Bird et al. (2014).



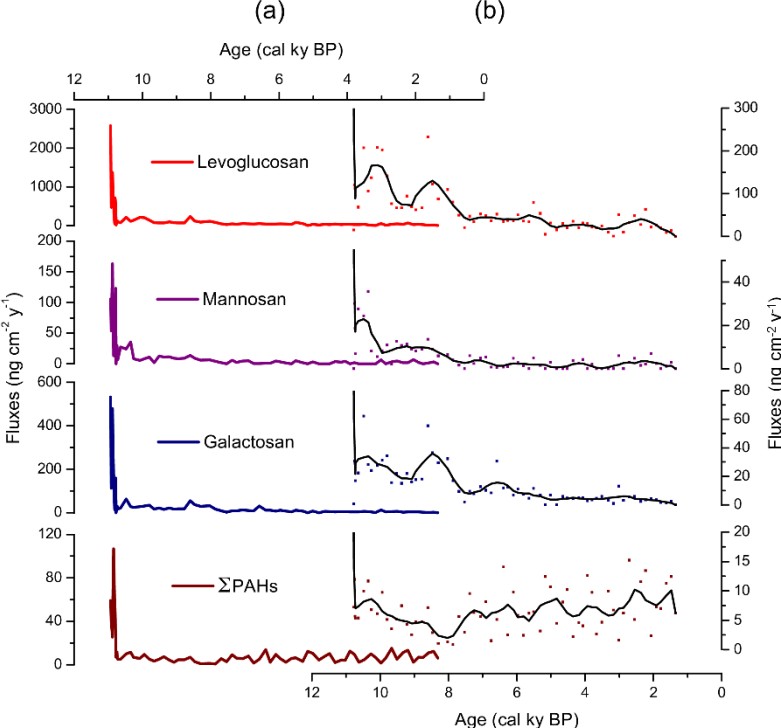

**Figure 2:** Fluxes of levoglucosan, mannosan, galactosan and PAHs in Paru Co (a) along the whole core and (b) zooming in after removing the deepest samples older than 10.7 cal ky BP (scatters and trends, obtained with 5-points weighted-moving averages).





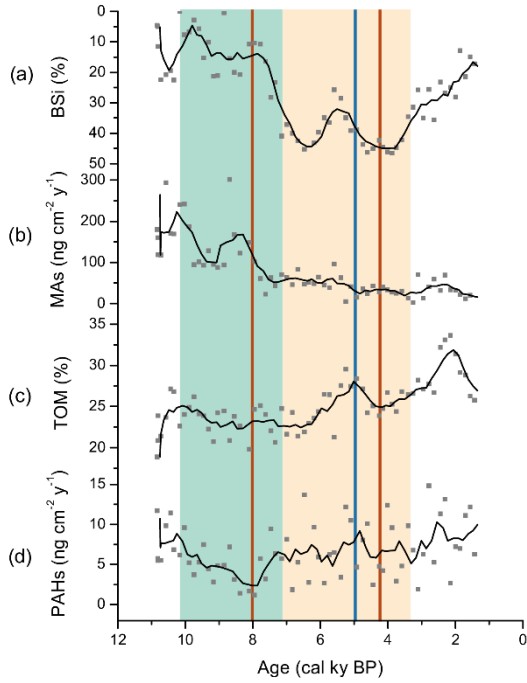

**Figure 3:** Results from Paru Co showing data and trends (5-points moving averages) (a) % BSi; (b) ΣMAs; (c) % TOM; (d) ΣPAHs. Blue box: ISM rainfall increased from 10.1 to 7.1 cal ky BP. Peach box: decrease of ISM rainfall to a minimum between 7.1 and 3.4 cal ky BP. Red lines: Bond events 5 (~ 8 cal ky) and 3 (4.2 cal ky). Blue line: division between warmer/wetter ISM and successive cooler/drier conditions.



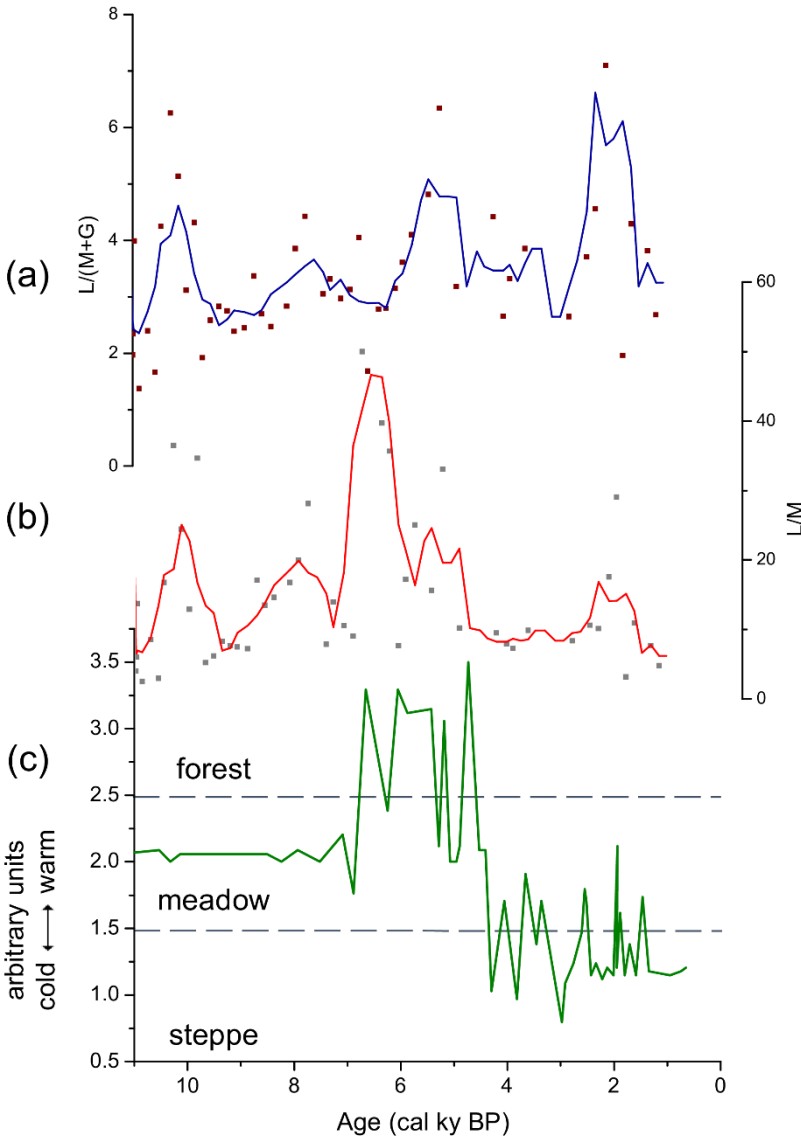

**Figure 4:** (a) Paru Co L/(M+G); (b) Paru Co L/M (trends obtained with a 5-points moving average); (c) Hidden Lake pollen inferred vegetation, data graphically obtained with GetData Graph Digitizer from Tang et al., (2000).




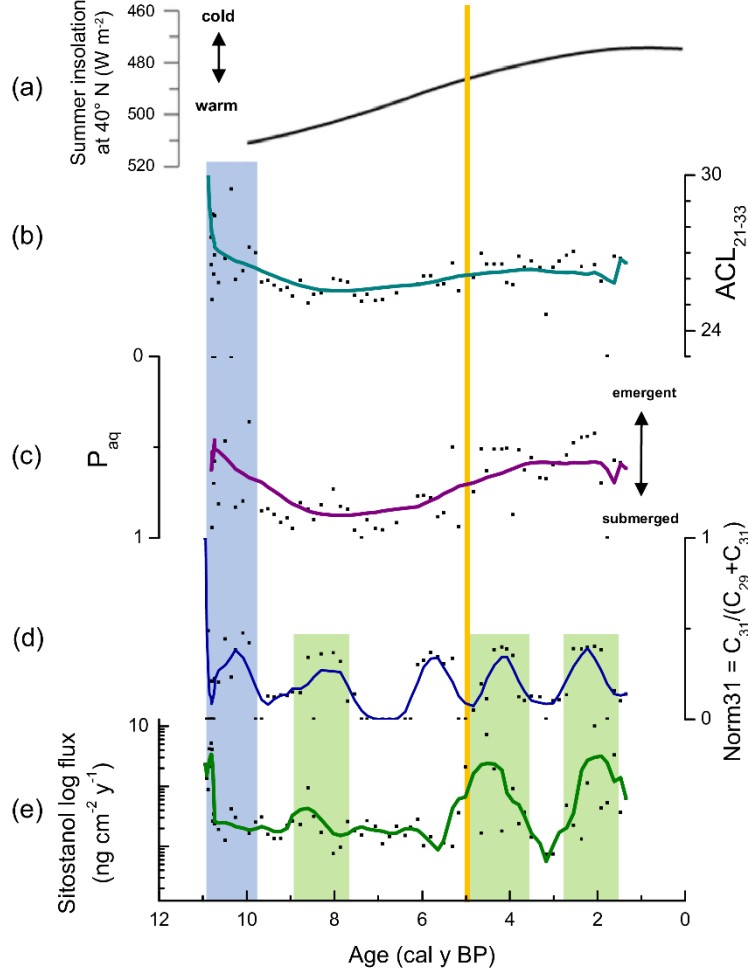

**Figure 5:** Comparison between vegetation indicators in Paru Co and association with the solar summer radiation. (a) summer insolation at 40° N (Berger and Loutre, 1991); (b) average chain length *n*-alkanes ratio; (c) $P_{aq}$ *n*-alkanes ratio; (d) Norm31 *n*-alkanes ratio; (e) Sitostanol log fluxes detected in Paru Co. Trends obtained with 5-points (Norm31 and sitostanol) and 10-points ($ACL_{21-33}$ and $P_{aq}$) weighted-moving averages. Gold line: division between warmer/wetter ISM and successive cooler/drier conditions. Blue bar: early Holocene variability of the proxies. Green bars: association between sitostanol and Norm31 oscillations.





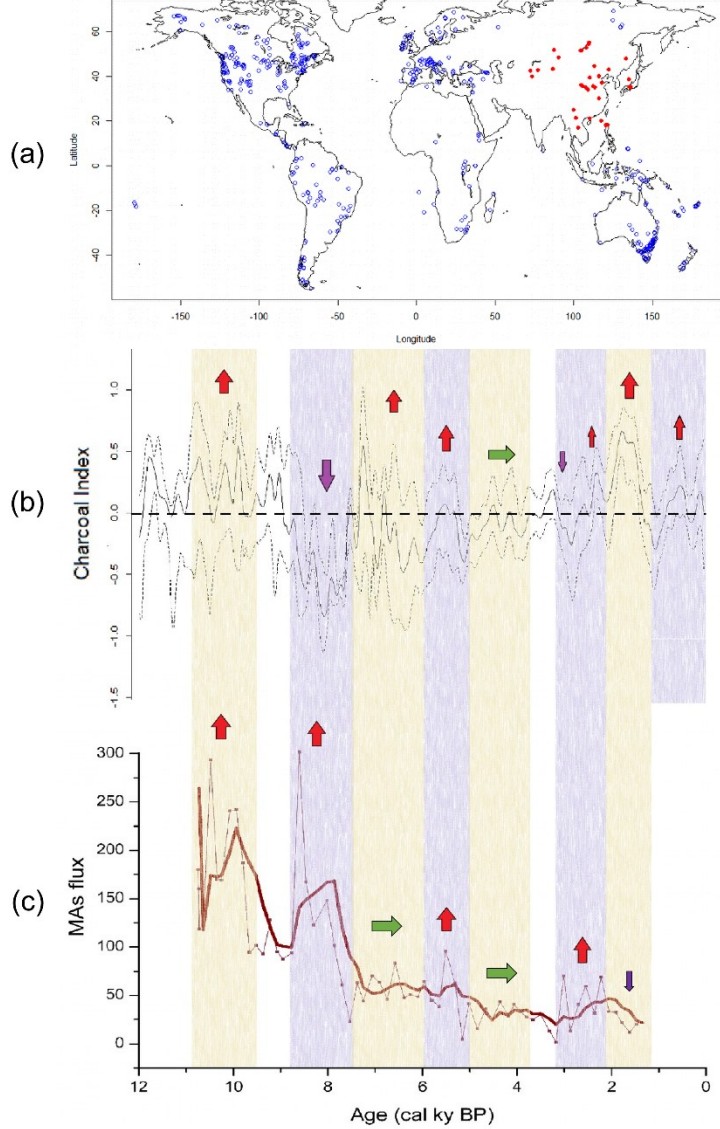

**Figure 6:** (a) Map indicating the 43 sites (red dots) used for the charcoal index elaboration; (b) charcoal index calculated with the GCD; (c) MAs (ng cm$^{-2}$ y$^{-1}$) data points and trend (5-points moving average) resulted from Paru Co analysis.





**Table 1:** Target molecules with their abbreviations, detected mass to charge ratio (m/z) and method detection limit (ng) calculated as blank values plus three standard deviations.

| Molecular classes | Compounds | Abbreviation | Targeted ion (m/z) | MDL (ng) |
|---|---|---|---|---|
| MAs | Levoglucosan | L | 161 | 226.1 |
| | Mannosan | M | 161 | 12.2 |
| | Galactosan | G | 161 | 9.8 |
| PAHs | Naphthalene | Naph | 128 | 16.0 |
| | Acenaphthylene | Acy | 152 | 8.4 |
| | Acenaphthene | Ace | 154 | 17.8 |
| | Fluorene | Flu | 166 | 8.7 |
| | Phenanthrene | Phe | 178 | 21.4 |
| | Anthracene | Ant | 178 | 19.3 |
| | Fluoranthene | Fluo | 202 | 11.1 |
| | Pyrene | Pyr | 202 | 19.4 |
| | Benzo(a)anthracene | BaAnt | 228 | 0 |
| | Chrysene | Chr | 228 | 0 |
| | Retene | Ret | 234 | 0 |
| | Benzo(b)fluoranthene | BbFl | 252 | 0 |
| | Benzo(k)fluoranthene | BkFl | 252 | 0 |
| | Benzo(a)pyrene | BaPyr | 252 | 15.8 |
| | Benzo(e)pyrene | BePyr | 252 | 0 |
| | Benzo(ghi)perylene | BghiPer | 276 | 0 |
| | Indeno(1,2,3-c,d)pyrene | IPyr | 276 | 0 |
| | Dibenzo(a,h)anthracene | DBahAnt | 278 | 0 |
| *n*-alkanes | $C_{10}$-$C_{35}$ | $C_{10}$-$C_{35}$ | 71 | 4735.0 |
| FeSts | Coprostanol | Cop | 215 | 0 |
| | Epicoprostanol | e-Cop | 215 | 0 |
| | Cholesterol | Chl | 370 | 100.4 |
| | Cholestanol | 5α-Ch | 355 | 0 |
| | Sitostanol | 5α-Sit | 215 | 0.5 |
| | Sitosterol | Sit | 215/396 | 2.3 |