# Peer review of "Fire, vegetation and Holocene climate in a south-eastern Tibetan lake: a multi-biomarker reconstruction from Paru Co"

_Climate of the Past, 2018_

## Referee Comment (RC1) · Anonymous Referee #1 · 5 Apr 2018

General comments

In this manuscript, A. Callegaro and colleagues present a multi-proxy investigation from a sediment core retrieved from a Lake on the Tibetan Plateau. They conducted relatively novel biomarker analysis to reconstruct past fire activity and vegetation in this area over the last 11 kyr.

The paper is relatively well written and structured, and addresses scientific questions relevant to the scope of Climate of the Past. However the presentation of the results (figure) and more importantly the discussion and argumentation need to be improved and strengthened. I will highlight several cases under my "specific comments – major

issues" where the argumentation was to superficial. Several possibilities for interpretation or explaining the discrepancies are often presented (e.g. fire activity, different transport, different fire temperature), which is good. But in the end the authors need to clearly state which one they favor and why. Looking at the figures, I mostly see discrepancies between the proxies. It may well be the case as the authors are comparing quite different indicators sometimes only marginally influenced by the parameter they are investigating (e.g. the effect of fire on n-alkanes). I strongly advise the authors not to go into too many directions, especially too many comparison with other proxies or records that do not match. But rather make sure they have good arguments for their interpretation in the end. I would rather recommend showing a record of ISM intensity or temperature over the Holocene.

The data are always presented with moving averages, which makes it difficult for the reader to appreciate for himself the original data obtained.

Specific comments Major issues:

- Page 9 line 2: I have to admit that I am quite skeptical about the high fluxes observed in both MAs and PAHs. As you state a few lines down, these high fluxes are the result of the higher sedimentation rate observed in the bottom part of the core. There is only one age point that is causing this high sedimentation rate, and no errors on this radiocarbon age are provided. Could it be that the sediment was distorted (stretched) during the coring process, causing these higher sedimentation rate. After checking Bird et al. (2014), it appears they dismissed a date which was much older (17.7 cal kyr) at 388.6 cm, as it was bracketed in between two ages in stratigraphic order. It could also be that your last one is a contamination during coring or bioturbation and that instead sedimentation rates were much lower in the bottom part of the core. Alternatively, the very high sedimentation rates could reflect an erosive event in the catchment. The TOM shows lower values for these older sediments. Erosion of catchment soil could bring also older fire biomarkers into the lake. Looking at the fluxes on figure 2b (ignoring the high peak), the MAs and PAHs don't show such similar patterns, except for the first

maximum around 10.5 kyr BP and the following decrease to 10-9 kyr BP. You do make this statement further down on lines 23-24. You never actually explain this difference. - Page 10 Line 25-26: You also never explain what could cause the difference between the 2 ratios L/M and L/(M+G), and which ratio is the more trustworthy, or which one you use for reconstructing the vegetation. - Page 11, GCD results: I don't quite follow your argumentation. Why did you compile charcoal records over such a vast area? Are you expecting similar climatic trends over the Holocene, over this entire area 1000s of km across? If yes, then you should make it clear why (monsoonal systems etc). If no, then it doesn't make much sense to compile all of these into one record. You should also discuss in more details the different temperature of production of charcoal and levoglucosan, what does it imply? What explains different fire temperature, and what fire would you then expect to explain your data. - Page 13: It would have been useful to show a record of changes in monsoon strength (e.g. precipitation) along your own records. - Page 13: If you would rather not trust your PAH record as a fire record, then you should make it much clearer earlier on, and mention that you will then only discuss MAs. Given the high variability of your PAHs, it may be your best option. - Page 14 line 7: Are you now argumenting that the MAs peak at 5.6 is due to transport, and not fire activity? A bit earlier you were discussing the Bond Event 4 and monsoonal precipitation. This is somewhat confusing. If both could play a role, you should add a summary sentences stating that.

Smaller issues:

- Title: I recommend adding "Lake" to Paru Co - Abstract, line 24: I would briefly explain why PAHs decreases but MAs remain high. What is the distinction to the intense biomass burning during the early Holocene where both were high? - Page 2, line 5: In the sentence just before you state that fire contribute to greenhouse gases, this contradicts the end of the sentence here. - Page 2, line 6: Why do you use "therefore"? I don't see a clear link into this last sentence. It makes sense that these sentences are in the introduction, but there is not much flow, or logical order to these sentences. They just

seem to be put together. Please improve the argumentation. - Page 2, line 13: (and throughout the manuscript) List references in chronological order, the oldest one first, the most recent one last) - Page 2 line 16: Can you specify what type of ecosystem processes? - Page 2, line 18: Ice cores from where, also the TP? - Page 2 line 26: I don't see what you mean by specific environmental conditions? - Page 2 line 31: I would clarify here that the following list of marker you are discussing are Mas - Page 4 line 25-26: Please rephrase this sentence: the first part of the sentence is about difficult access to paleoclimate archives and then you mention few investigations into species diversity and plant communities. Where is the relationship? - Page 5 line 19: Define ecosystem functions or use another word, e.g. vegetation distribution? - Page 5 and Figure 1: It would have been more useful to have a more precise catchment map showing these features thant the large google map on figure 1 or the satellite picture showing only the lake. - Page 6 Line 8: The 137Cs determination method is not cited here, whereas the radiocarbon is. - Page 8 line 7: How did you obtain wet density? - Page 8 line 11: for the other ratios you clearly state what they are useful for. You should do the same with the ACL, what can it tell you? - Page 8 line 29: please provide these latitude an longitude ranges. How many records did you compile in total? - Page 9 line 7: or you had erosion of older compounds in the catchment (soil). - Page 9 line 30: Why didn't you look at the correlation between BiSi and PAHs and TOM and PAHs? - Page 9 line 33: It would have been good to summarise here how the link ISM-BiSi works. - Page 10 line 1-9: These correlations should be presented when you first describe similar trends of both PAHs and MAs, at the beginning of the section! - Page 10 line 2: the (negative) correlation between MAs and TOM was larger than this (-0.54) and with a more significant p-value. Maybe there is something to discuss there, even though it is not positive as you expected. - Page 10 line 5: how does it vary? Does it vary with time, depending on the main climate? - Page 10 lines 5-9: These last two sentences should probably be moved to the discussion section. - Page 10 line 7: Unfortunately we don't see the original concentration in your figures, only the fluxes. - Page 10 line 8: can you specify what you mean by "biogenic origin"? combustion of biomass is also biogenic

for me - Page 10 Line 13-15: I don't fully understand this sentence, can you rephrase it to be more clear? - Page 10 line 15: Is the statistic done only for this interval, or for the entire core? please specify - Page 10 line 19: please provide some examples of these changes in the terrigenous environment. - Page 10 line 28: It would be useful to either state the published range in the text, or in the figure. Are those ranges for L/M, or for L/(M+G) - Page 11 line 1: the second part of this sentence is rather vague. n-alkanes do not record all organic input inot the lake, and they also record organic production within the lake. - Page 11 line 15: Here you should list the other FeSts, especially those that would have indicated the presence of humans. - Page 11 Line 16-17: Are there other information (e.g. archeology) which could support this finding? No known settlement in this area, too high elevation, ...? - Page 11 lines 19-23: All this first part should be in the method section. - Page 11 line 31: I only counted 3 colour bars where the arrows go in the same direction, that's pretty bad as a similarity... - Page 12 line 31: I wouldn't call this composite record regional, it's almost continental - Page 13 line 5: How do these different burning temperature occur? You need to discuss this point in further details. Would we then have low or high temperature fire during this interval, and why, what caused this type of fire? - Page 13 line 14: The PAHs show a clear minimum at 8kyr. The Sum of MAs show a peak from 8 to 9 kyr BP. The 8.2 cal ky BP was a short and abrupt event, if the ISM was peaking then, I would not expect a 1000 year long dry interval. - Page 15 line 4: that warm period would fall right into the 8.1 to 7.2 cold intervals (1-2 degrees cooling) you are mentioning a few lines up... that's contradictory - Page 15 lines 6-7: please specify the time interval here. As you were just mentioning relatively young intervals (<2.7 kyr BP) where you don't have fire records it is confusing. Do you mean for the early Holocene? - Page 15 line 18-19: I wonder which one (forest or shrub) tend to have more fire? I also wonder if the presence of forest or shrub has an influence on the fire temperature? - Page 15 line 22-24: In this sentence is is hard to follow what the observations where, and what are the suppositions, could you reformulate more clearly what has been observed, and what is assumed? - Page 15 line 32-33: I don't follow your argumentation. How can you infer

that this mechanism would also occur in the sediments? - Page 16 line 1-2: leaf waxes can also be abraded and transported by the wind, as well as in streams in suspended sediments, leaves are not necessarily requested for their transport and deposition - Page 16 line 6: I don't understand what you insinuate here? What details would you look into? Please be more specific and/or provide examples. - Page 16 line 25: No, they can also be transported quite far by winds. - Page 17 line 1: what do you mean by paleoreconstruction information, this is too vague, could be paleotemperature, paleo-precipitation... - Page 17 line 27: You should not discuss something for the first time in the conclusion. This expansion of Bronze Age civilization should have been mentioned earlier. In the text, you mention 4.2k as the collapse of Chinese Neolithic cultures - Page 31 – figure 5: Your lake is located at 30N, why do you use the insolation at 40N? - Page 31 – figure 5: There are dots in between the Paq and the ACL graphs, to which graph do they belong to? - Page 31 – figure 5: I wonder what signal you would obtain if you were to use the same 5pt moving average on the Paq and the ACL. It seems to me that the Paq data without moving average show a signal similar to the sitostanol - Page 31 – figure 5. I am not quite convinced by the insolation driven trend. Or at least I don't think this is the signal you should be looking for in your n-alkanes data over the Holocene. I am also not convinced by your green bars highlighting similar oscillations in Norm 31 and sitostanol. The youngest peak is relatively coeval, the one before is already almost opposite (sitostanol peak is closer to the Norm31 minimum than to its maximum). The third Norm31 peak is not coeval with any peak in Sitostanol. And the 4th peaks are again quite offset. - Page 32 Figure 6: The figure is of poor resolution. The blue dots on the map in panel a are not described/explained. If not used these should be removed. The oscillations in panel b (charcoal) can barely be seen. The curve and its envelope (dotted curves) are not explained. I suggest deleting "resulted" and "analysis" from the figure caption. The figure caption should also describe what the blue green and red arrows are for.

Technical corrections - Page 2, line 11: delete provided - Pag 2 line 28: delete "in buried sediments" - Page 2 line 31: You could add "and longer timescales" after "the

Holocene" - Page 3 line 17: I would here mention "diverse distribution of chain length" - Page 4 line 1: replace "anthropological" with "archeological"? - Pag 4 line 2: replace "quantification" with "determination" (I wouldn't say that we can truly quantify the presence of humans. We are not there yet. - Page 5 line 5: You could mention the Younger Dryas & Bolling Alerod in your text. - Page 5 line 12: You could maybe use "superimposed on these oscillations" instead of "even with these oscillations". - Page 8 line 3: I would rather use data "analysis" instead of "elaboration" - Page 8 line 9: Replace "significative" with "significant" - Page 8 line 11: what do you mean with "useful for work"? I would rather say "useful for our study, or for our interpretation" - Page 8 line 30: Here you should refer to the figure presenting this data - Page 9 line 17: use "up to" instead of "touching values till" - Page 9 line 21: remove the comma - Page 13 line 27: delete the second "not" - Page 15 line 2: Is there a word missing? ("a limited abrupt to"???) - Page 15 line 13: I would rather use another verb, for instance "place the Paru Co..." - Page 15 line 15: specify the time interval considered - Page 15 line 26: I would rather call this "long term trend" than "millenial scale". - Page 15 line 29: can you indicate by how much on average? - Page 15 line 30: same here, by how much? is it significant? - Page 16 line 4: "it seems that" - Page 16 line 11: slash and burn -> at which time? - Page 16 line 11: Please indicate Lhasa on Figure 1. - Page 16 line 12: where exactly? Please show on the map. - Page 16 line 16: compared to (or "with respect to") - Page 16 line 32: Comparing data that resulted from (or originated from) - Page 16 line 33: "processes that happened" - Page 19 line 6: The journal and pages are missing. - Page 29: Specify in the figure caption that the BSi axis is inversed

---

## Short Comment (SC1) · 7 Apr 2018

The PAGES Data Stewardship Integrative Activity seeks to advance best practices for sharing the data generated and assembled as part of all PAGES-related activities. The CP Special Issue, "PAGES Young Scientists Meeting 2017" is part of this PAGES activity. The co-editors of the Special Issue are reviewing the data availability within each of the CP-Discussion papers in relation to the CP data policy (https://www.climate-of-the-past.net/about/data_policy.html) and current best practices. The editor team is making recommendations for each paper, with the goal of achieving a high and consistent level of data stewardship across the Special Issue. We recognize that an additional effort

will likely be required to meet the high level of data stewardship envisaged, and we appreciate the dedication and contribution of the authors. This includes the use of Data Citations (see example below). Authors are also strongly encouraged to deposit significant code into a suitable repository and to cite it using a Data Citation.

We ask authors to respond to our comments as part of the regular open interactive discussion. If you have any questions about PAGES Data Stewardship principles, please contact any of us directly.

Best wishes for the success of your paper.

YSM Special Issue editor team (E. Dearing Crampton-Flood, D.S. Kaufman, R. Barnett, M.F. Loutre, M.N. Evans, S.C. Fritz, C. Tabor, Y. Zhang, E. Razanatsoa, and H. Plumpton)

For this paper:

All papers submitted to Climate of the Past must include a Data Availability section that details the location of the data that were used as input to the study, including previously published data that were used for comparison purposes, and the data that were generated by the study.

(1) Research input data-

The paper makes use of the Global Charcoal Database (GCD). In order to adhere to the Data Policy for Climate of the Past, persistent identifiers (doi or URL from NOAA Paleoclimatology), or full data citations to the primary data must be included in the Data Availability section.

Fig. 1 (c) includes data from the TRMM dataset, which is not in the reference section. Please add the appropriate data citation to the paper.

Fig. 3 includes proxy data from Bird et al. (2014). We were delighted to discover that the data are already available in a public repository at

Interactive
comment

https://www.ncdc.noaa.gov/paleo/study/16399. Add a data citation for this dataset to the figure caption and to the Data Availability section.

Fig. 4 (c) includes a summary of pollen data from Tang et al. (2000) that was digitized from the original publication. Digitizing data from previous publications is a legitimate practice, but has important disadvantages. The accuracy is often degraded by the digitizing, and without essential metadata, the data are not available for reuse. Instead of digitizing previously published data, we encourage authors of this special issue to serve as data stewards by working with data generators to rescue and properly curate important datasets that are used in their paper. Please contact Tang to explain the re-use of the data and to offer to facilitate the transfer of the dataset to a data repository. Once deposited in a repository, the data can be cited with a data citation.

Fig. 6. Include a data citation to the GCD in this figure caption and in the Data Availability section.

(2) Research output data –

Biomarker data plotted in Figs 4a-b, 5b-e, 6c: Monosaccharide anhydrides (MAs), polycyclic aromatic hydrocarbons (PAHs), n-alkanes, fecal sterols and stanols (FeSts) – This paper presents new and valuable biomarker data for the south-eastern Tibetan Plateau during the Holocene. These new data must be uploaded to a long-standing online data repository, and a data citation or URL link from NOAA Paleoclimatology for access to these data must be provided in the Data Availability section of the paper.

Charcoal index plotted in Fig. 6b: Composite of previously published data from 43 sites – This is an important new summary of the regional fire history based on available charcoal records. The outcome of the synthesis (the time series with uncertainties) should be transferred to a repository along with a table of metadata that includes the name and location and reference for each of the 43 sites. This product could be included on the same landing page, with the same doi/NOAA URL along with the biomarker data.

What is a "Data Citation"?

Data Citations track the provenance of a dataset giving credit to the data generator; this is in addition to any references to publications where the data are described. Data Citations are used in the text (or tables) alongside and in the same way as publication citations. In the Reference list, they include: Creators, Title, Repository, Identifier, Submission Year. More information about Data Citations is here: <https://www.datacite.org/mission.html>

Here is an example of text and corresponding citations (using CP punctuation style):

"The PAGES2k Consortium (2017a) assembled a large global dataset of temperature-sensitive proxy records (PAGES2k Consortium, 2017b). Among the records is the paleo-temperature reconstruction from Laguna Chepical (de Jong et al., 2016), which was described by de Jong et al. (2013)."

References

de Jong, R., von Gunten, l., Maldonado, A., and Grosjean, M.: Late Holocene summer temperatures in the central Andes reconstructed from the sediments of high-elevation Laguna Chepical, Chile (32° S), Climate of the Past, 9, 1921-1932, 2013.

de Jong, R., von Gunten, l., Maldonado, A., and Grosjean, M.: Laguna Chepical summer temperature reconstruction, World Data Center for Paleoclimatology, https://www.ncdc.noaa.gov/paleo/study/20366, 2016.

PAGES 2k Consortium: A global multiproxy database for temperature reconstructions of the Common Era, Scientific Data, 4,170088, 2017a.

PAGES 2k Consortium: A global multiproxy database for temperature reconstructions of the Common Era, version 2.0.0, figshare, https://figshare.com/s/d327a0367bb908a4c4f2, 2017b.

---

## Referee Comment (RC2) · Anonymous Referee #2 · 9 Apr 2018

General Comments: In this paper Callergaro et al. present results from a biomarker multiproxy reconstruction of fire and vegetation from lake sediments Holocene on the Tibetan Plateau. The methodology used in this paper and the scientific aims of this study are will within the scope of this journal. This paper applies a clever approach where multiple lines of environmental evidence (i.e. fire, vegetation, human/animal habitation) can be reconstructed from the same samples using a relatively streamlined workflow. Additionally, I appreciate the authors' tactic of using data from the GCD to interpret their fire data within a regional framework.

Despite these strengths, this paper could be improved by better presentation (figures)

[Figure]

and clearer interpretation of the data. I found myself unable to follow the logic at times, and occasionally, the data and interpretations seemed at odds with each other. Adding more complete explanations of proxy interpretations (in both the text and the figures) may clear up some of this confusion. There were also times in the discussion where evidence from other studies was presented without being linked to the new data, and the new conclusions felt buried. Make sure to emphasize the novel contributions of your work and what it adds to the literature framework.

I have outlined some more specific issues below. Making these improvements will greatly increase the readability of this paper and strengthen the arguments.

Specific Comments: Page 8 Line 31: Are you using %BSi as a proxy for monsoon intensity? If so an added sentence explaining why would be helpful. Also did you measure %BSi or %TOM or is it from Bird et al 2014? Please specify.

Page 9 Lines 13: I don't know if concurrent increases between PAHs and TOM implies a specifically biogenic origin for PAHs, just that the total organics in the lake and the PAHs may have a similar source. Especially given that your aquatic/terrestrial indicators show an increase in terrestrial n-alkanes to the lake after 8 cal ky. Perhaps there is more windblown terrestrial material being added to the organic pool and that's why it's increasing? You touch on this in the next paragraph.

Page 9 line 20: MAs are more water-soluble than PAHs, so this argument doesn't make a lot of sense

Page 9 Lines 31-33: You need to explain and cite how you are interpreting these ratios and what is the difference between L/M and L/(M+G).

Page 10 Paragraph on Line 5: ACL and Paq represent indices for differential terrestrial/aquatic inputs I'm not sure how that directly relates to interpretations of fire and vegetation change. Make it clear if you are relating this to lake levels and climate because these proxies don't explicitly address changes in terrestrial vegetation community.

Page 10 Line 19: Are you using this to say something about source area changes or vegetation community changes? This distinction is not clear.

Page 10 Line 23: Is there any evidence that sitostanol is correlated with grassy tissues? Citation?

Page 11 Lines 13-32: This paragraph is under "paleofire activity" but doesn't mention fire at all, just gives some climate context. Perhaps having a climate section would be useful?

Page 12 Lines 9-15: Throughout the manuscript, potential reasons for the differences between the MA and PAH records are mentioned, but no evidence is presented to support any of these interpretations over the other. Are there sedimentary changes (i.e. grain size) when these records diverge that indicate changes in transport to the lake? Are there changes in PAH ratios throughout the core that might indicate changes in transport/fire temperature? Perhaps there are inherent differences in the transport of MAs and PAHs due to their size/solubility differences? Why might the charcoal and PAH record correspond better than the charcoal and MA records? If these questions are explicitly addressed it will greatly strengthen interpretations of fire history. If you haven't already, I'd suggest reading Denis et al. 2012 Organic Geochemistry, which has a good discussion considering transportation/degradation/fire temperature differences in lake fire proxy records.

Page 12 Line 26-28: What particular climate effect would Bond events have on fire in Tibet? Link through a mechanism

Page 13 Lines 1-4: Like I said in the results, I don't think you can assume that the PAHs are biogenic. Not without some additional evidence. Have you tried to use some PAH degradation ratios? The two papers you cite here are good resources for these tools. I'd be interested to see if applying the appropriate ratios to your data provide evidence

for degradation.

Page 13 Line 5-15: I appreciate bringing in this discussion of transport. . .but I feel like how this relates specifically to your data gets lost. Does the peak at 5.6 represent increased wind, fire or both?

Page 13 Lines 14-15: Are there dust layers recorded at Paru Co?

Page 14 Lines 4-13: The times you highlight as having higher vegetation density are not all correlated with your fire records. Make sure you state when this mechanism applies and when it does not, and why that may be.

Page 15: I think that you need to address explicitly how the vegetation changes (i.e. shrubs versus trees) that are observed in the record are (or aren't) tied to your record of fire activity.

Page 15 Lines 2-6: Leaves deposited into the lake are not the only source of terrestrial alkanes, they can also be ablated off leaves and transported by wind.

Figure 3. Instead of color boxes interpreting the ISM instead show the original proxy record you are using for those interpretations. The $\delta$Dalkane record from this same core from Bird et al. 2014 would be perfect to show plotted against proxies that you argue are influenced by ISM variations.

Figure 4. A and B could use some interpretive annotations. What to high/low MA ratios mean in terms of vegetation community? This is something that is unclear throughout the entire manuscript. How are these ratios interpreted and why do they differ? Adding some interpretive lines (like in 4C) would be very helpful. 4C. What is meant by arbitrary units? Explain this in the caption.

Figure 5. I think plotting these records with a moving average obscures some important variability. It almost seems that the ACL and Paq have millennial cycles. Looking back at Bird et al. 2014, this seems to vary with reconstructed lake levels, rather than insolation as you argue. I think plotting that data in the same figure as these would

really strengthen the interpretability of these proxies. Also, ACL and Paq and Norm31 and Sitostanol are really proxies for different things (aquatic v. terrestrial vegetation and changes in terrestrial community). It would be much clearer to separate these proxies into different figures. Perhaps, one figure with lake levels, insolation, ACL and Paq, and then add Norm31 and Sitostanol to Figure 4, which is your terrestrial vegetation figure.

Figure 6: Pg. 10 line 33 it is stated that the charcoal data from the GCD was drawn from a 1000 km radius from the Paru Co site. . .However, after a quick check on Google maps, I found all of the red dots on the Fig 6a map are actually more that 1000 km away from the site. Please address this contradiction and correct how the GCD sites were chosen in the text. Additionally, this large area of integrated charcoal records is potentially problematic because this data is drawn from a large continental area and the assumption that these sites are subject to the same climate conditions as the Paru Co site may no longer hold.

Figure 6 B and C: The shaded regions and arrows are not explained in the caption. And if they are showing correspondence then it almost seems that these records hardly reflect each other (which is discussed a bit in the text). Why is the PAH record not shown? I know it is highly variable, but it actually may correspond better to the charcoal record. . .if this is the case, then what are the implications for your data?

Technical Corrections: Page 2 Line 1: The dependent clause of this sentence is unclear

Page 2 line 13-15: perhaps an i.e. style list of just three or so methods

Page 2 Line : move "in the last Century" to after "biomass burning"

Page 4 Line 6: Perhaps this would be a good place to introduce Neolithic/bronze age societies you talk about in the discussion.

Page 4 Line 13: cite Figure 1a

Page 5 Line 30: cite Figure 1d

Page 6 Line 18: new paragraph at "Each Sample"

Page 7 Lines 18-19: What company did you obtain your standards from?

Page 8 Line 11-12: This topic sentence does not fit the content of the paragraph. Additionally, I think you actually end up arguing that these records don't agree?

Page 9 Lines 10-13: This sentence is confusing and potentially unnecessary since you elaborate on it in the results.

Page 10-11 Section 4.3: This reads like a methods section rather than results. Move this to methods and instead describe the trends you see in your analysis in the results.

Page 11 Line 29: delete "a" before decreasing

Page 12 Lines 15-19: This is a rambling sentence; perhaps splitting it into two would make it clearer?

Page 15 Line 6: you seem to be missing a word after sedimentological

Page 15 Line 8: delete "seems that and add "also" between "could" and "help"

Page 15 Line 26: I'm confused what you mean with "except when associated to PAHs"

Page 15 Line 28: Citation needed Bush and McInerney (2013) GCA and/or Diefendorf et al (2011) GCA.

Page 15 Line 27: Be consistent with the use of pollens or pollen, don't switch off

Page 16 Line 28: I think this is the first time you mention Bronze Age civilizations. You should elaborate on this earlier in the paper.

Page 16 Line 32: This is an abrupt way to end. Perhaps add a sentence of significance or implications?

Figure 1. Include what the dates in 1D are based on (14C?).

[Figure]

---

## Referee Comment (RC3) · Anonymous Referee #3 · 12 Apr 2018

General Comments:

In this manuscript, Callergaro et al. demonstrate the usefulness of a multi-proxy approach to reconstruct fire and vegetation change throughout the Holocene from a lake sediment record in the Tibetan Plateau. The research objectives and methodology used in this study are within this journal's scope. This manuscript attempts to reconstruct fire, vegetation change, and human presence nearby the lake using biomarkers, and discuss how the results and other regional analyses of fire and climate compare.

This manuscript presents a unique and novel record for the TP region. However, this paper could be improved with better interpretation of data, as well as better figures

and presentation of data. The data and the author's interpretation seemed to conflict with eachother, particularly with the fire records presented. Adding more background information, discussion, and analysis of some of the records (detailed below) could drastically improve this paper and the discussion of results. I have presented some more specific issues below, listed from "Major Comments" to "Minor Comments" to "Figure Comments". Making these improvements will greatly increase the readability of this paper and strengthen the arguments.

Major Comments:

I'm not sure I follow the PAH argument. In the manuscript, you argue that PAH track local and regional fire activity, along with MAs, in the early portion of the record (10.7-8.7), but may switch to having biogenic origins after 8.7, just due to the fact that they correlate slightly with TOM. Looking at Fig 3, it seems as though the PAHs are actually making sense as a fire proxy more-so than MAs- lower values during the times of increased ISM rainfall, higher values during the times of decreased ISM rainfall. Furthermore, the "noise" in the PAH record looks like millennial scale fluctuations in the fire activity, which you aren't capturing in the MA records. I would suggest more discussion on the PAHs as potentially tracking fire activity, instead of just writing them off as being biogenic in nature. There are many different ratios of PAHs that studies have shown to prove useful in determining PAH source (i.e. biomass burning vs fossil fuel burning, biogenic vs burning, etc. . .). Possibly look into some of these ratios as well to see if you can determine a ratio that is suitable for developing your story. Some papers that use ratios include:

Denis et al. (2012). PAHs in lake sediments record historic fire events: Validation using HPLC-fluorescence detection. Org Geochem.

Miller et al. (2017). Local and Regional Wildfire Activity in Central Maine (USA) during the past 900 years. Journal of Paleolimnology

Yunker et al. (2002). Sources and significance of alkane and PAH hydrocarbons in

Canadian arctic rivers. Estuar Coast Shelf Sci

I'd like to note that just through comparing Figures 2 and 6 by eye, it seems like the PAH record tracks the GCD regional composite record fairly well (at least way better than the MA record does). I would advise plotting the PAH curve on figure 6 – that way we can visualize how the PAH record tracks regional fire activity.

Minor Comments:

Page 13, line 23-24: "In general, fire history shows a decreasing trend from 8 cal ky BP to the present" – this isn't apparent based on the figures you show. The MAs decrease, but the PAHs steadily increase. Distinguish between the two instead of saying "the Paru Co fire history"

Page 17, line 7: this should be labeled the MA fire history, not the overall fire history from your record. The PAH fire history shows the opposite of this – with lowest values at 8 cal kyr BP, and then a long term increasing trend.

It could be beneficial to include coring location in lake and a bathymetric profile of the lake. Looking at lake bathymetry could give insight or possibly explain some of the trends seen in the data, and could manipulate the age-to-depth model so that it isn't linear in reality.

Please check target ions for each compound – for example, many studies that look at retene have a target ion of 219 instead of 234 (the compound's molecular weight). The mass spectra can be found here: https://webbook.nist.gov/cgi/cbook.cgi?ID=C483658&Units=SI&Mask=2380#IR-Spec. Using 234 may be adequate, but in some cases using non-major ions may "hide" compounds, particularly when running in SIM mode on a GC-MS. In your case, it seems that this may in fact be occurring, since you report retene was undetectable in most/all samples. Given the fact that retene is produced by combustion of coniferous trees, its surprising that retene is not found, given the fact that you mention a coniferous forest

near the lake (line 24, page 5).

The spikes seen in all data at the beginning of the record has me skeptical of whether or not it is a true signal of some climate/environmental variability. Adding in some discussion (1-2 paragraphs) on other, more plausible causes of this (i.e. an event in catchment that was preserved in the sediment record, a coring artifact, etc. . .) could give your arguments more validity throughout the manuscript.

Figure Comments:

Fig 1 a) the map seems a more complex than is necessary. The surrounding areas may not be as important to this study as the TP, so one option could be zooming in on the study region. One option could be to make it similar to the map in the supplement – that map is simpler and much easier to read, and having a map similar to that could more easily highlight the study areas in this figure. Also, you might want to confirm with google about publishing google map images in academic journals – I'm unsure if there are any special permissions needed from Google, but it could be a good thing to check.

Fig 2) do not overlap a) and b). This makes it seem as though there is a peak in values occurring at 4 cal kyr BP. There are multiple ways to fix this – you can either separate them so they don't overlap, or possibly highlight/box the areas in a) that are being zoomed in on in fig b).

Fig 3) try moving a) and c) y axes over to the right side – that way the axes are not overlapping or too close together.

Figs 2 and 5) use the same color between these two plots for similar things. For example, in figure 5 you use a gold line to separate ISM changes, while in figure 2 it is a blue line. Try to stay consistent in color schemes for the reader.

Fig 6) needs to be higher resolution. On figures b and c, you can barely see the lines. Making the lines bolded/bigger, as well as saving a high resolution image, would help

fix this issue. Furthermore, adding the PAH record, not just the MA record, would be very beneficial, as the PAH and GCD records seem to track eachother.

---

## Referee Comment (RC4) · J. L. Toney (Referee) · 24 Apr 2018

The manuscript "Fire, vegetation and Holocene climate in the south-eastern Tibetan Plateau: a multi-biomarker reconstruction from Paru Co" uses a suite of biomarkers to assess vegetation and fire change during the Holocene from a sedimentary record of a small lake on the Tibetan Plateau. This study presents original data with potentially interesting new results on fire history that has not been widely studied on the Tibetan Plateau.

The manuscript is well written and methodologies are robust. Given that this is a relatively new field of research, mainly the application of fire-related biomarkers to paleo-

climate records, there are additional aspects that the authors should consider.

PAHs: For instance, although they suggest that there are only a few studies of PAHs as tracers of biomass burning and only cite two (Page 3, Line 12), there are others out there that may help with their interpretations, for instance: Page et al. 1999, Marine Pollution Bulletin; Yunker et al. 2002, Organic Geochemistry; Denis et al. 2012, Organic Geochemistry; Yan et al. 2014, Environmental Toxicity and Chemistry; Yunker et al. 2015, Organic Geochemistry; and Denis et al. 2017, Organic Geochemistry. In particular, not all PAHs result from biomass burning, so using the sum of PAHs, for example, may not be as useful as targeting the pyrogenic PAHs (examples in Page et al. - fluoranthene, phenanthrene, benzo(e)pyrene). Denis et al. 2012 suggest that there are differences in high molecular weight PAHs representing the intensity of the fire (also see McGrath et al. 2003, Journal of Analytical and Applied Pyrolysis), whereas, the low molecular weight PAHs more consistently record local fire events. These considerations may or may not be applicable, but could be tested without acquiring more data. This analysis may help to resolve differences in between the MAs and the PAHs. Finally, with respect to the PAHs (Page 13, Line 28), if there is a change in the biogenic/diagenic signal of the PAHs, then it would likely manifest specifically in the PAHs like perylene - it would be worth having a look at how the individual PAH profiles change when this signal becomes prominent. Degradation, if it is of the overall organic material should also manifest in changes in the carbon preference index (CPI) values, but these data are not plotted. If a low CPI is seen during times of highly charred material, this index could help support the argument made on Page 16, Line 3.

n-Alkanes: It is worth applying some caution in the use of the Paq from Ficken et al. 2000, which was derived in from Mt. Kenya in Africa. The organic geochemistry community is finding that a site-specific approach may be needed and while the assertions about long-chain and short-chain n-alkanes generally hold true, in some environments the relationship is slightly more complex. For example, in Garcia-Alix et al. 2018, Scientific Reports, the supplemental information shows how this index and the ACL

vary with distance from water source. Because grasses are prominent during more humid conditions in the arid Sierra Nevada region, the C31 shows aquatic rather than terrestrial-type vegetation changes. This may apply to similar high-elevation sites on the Tibetan Plateau and should be discussed. It is not a fault of the authors, just a really recent paper that might change the interpretations made here. This could help explain why the n-alkanes are showing a different pattern of change than the MAs and the grass/wood prevalence of pollen data (Page 16, Line 27).

Overall, this is a very interesting and well thought out study, but further analysis given the above comments could help with discussion.

---

## Author Comment (AC1) · 9 Jul 2018

Anonymous Referee #1 General comments In this manuscript, A. Callegaro and colleagues present a multi-proxy investigation from a sediment core retrieved from a Lake on the Tibetan Plateau. They conducted relatively novel biomarker analysis to reconstruct past fire activity and vegetation in this area over the last 11 kyr. The paper is relatively well written and structured, and addresses scientific questions relevant to the scope of Climate of the Past. However the presentation of the results (figure) and more importantly the discussion and argumentation need to be improved and strengthened. I will highlight several cases under my "specific comments – major issues" where the

argumentation was to superficial. Several possibilities for interpretation or explaining the discrepancies are often presented (e.g. fire activity, different transport, different fire temperature), which is good. But in the end the authors need to clearly state which one they favor and why. Looking at the figures, I mostly see discrepancies between the proxies. It may well be the case as the authors are comparing quite different indicators sometimes only marginally influenced by the parameter they are investigating (e.g. the effect of fire on n-alkanes). I strongly advise the authors not to go into too many directions, especially too many comparison with other proxies or records that do not match. But rather make sure they have good arguments for their interpretation in the end. I would rather recommend showing a record of ISM intensity or temperature over the Holocene. The data are always presented with moving averages, which makes it difficult for the reader to appreciate for himself the original data obtained.

A: We thank Anonymous Referee #1 for useful comments that helped us improving the quality of the work. We substantially revised and rewrote great part of the paper following your indications. We reply to specific comments below.

Specific comments Major issues: Page 9 line 2: I have to admit that I am quite skeptical about the high fluxes observed in both MAs and PAHs. As you state a few lines down, these high fluxes are the result of the higher sedimentation rate observed in the bottom part of the core. There is only one age point that is causing this high sedimentation rate, and no errors on this radiocarbon age are provided. Could it be that the sediment was distorted (stretched) during the coring process, causing these higher sedimentation rate. After checking Bird et al. (2014), it appears they dismissed a date which was much older (17.7 cal kyr) at 388.6 cm, as it was bracketed in between two ages in stratigraphic order. It could also be that your last one is a contamination during coring or bioturbation and that instead sedimentation rates were much lower in the bottom part of the core. Alternatively, the very high sedimentation rates could reflect an erosive event in the catchment. The TOM shows lower values for these older sediments. Erosion of catchment soil could bring also older fire biomarkers into the lake. Looking at the

fluxes on figure 2b (ignoring the high peak), the MAs and PAHs don't show such similar patterns, except for the first maximum around 10.5 kyr BP and the following decrease to 10-9 kyr BP. You do make this statement further down on lines 23-24. You never actually explain this difference.

A: We agree with you the fact that the high sedimentation rate found in the deepest part of the core could have been derived by a distortion during the coring process or bioturbation, causing these higher fluxes of biomarkers. Due to this high uncertainty, we decided to discuss our biomarker's dataset only until 10.78 cal ky BP, as we state in our revised paper (P.6 L.12-14): "Since the deepest part of the core shows much higher sedimentation rate that cannot be clearly explained, with the possibility of data distortion, the subsequent description and discussion of the results exclude the samples aging 10.784-10.937 cal ky BP, limiting the dataset interpretation to the period between 1.347 and 10.768 cal ky BP". Moreover, as suggested by Anonymous Referee #3 we reanalyzed the samples for PAHs, including new target ions. The obtained results are different from the discussion paper and are shown in figure 2, that strongly changed.

Page 10 Line 25-26: You also never explain what could cause the difference between the 2 ratios L/M and L/(M+G), and which ratio is the more trustworthy, or which one you use for reconstructing the vegetation.

A: The difference between the 2 ratios is the inclusion of galactosan in the calculation. However, due to the fact that galactosan seems to show a different degradation pattern, we decided to discuss only L/M, as we state in the new version of the paper (P.10 L.2-4): "Due to the fact that galactosan presents a different biodegradation behaviour, the application of L/(M+G) ratio may be inadequate (Kirchgeorg, 2015). For this reason, we limited the discussion only to L/M ratio results."

Page 11, GCD results: I don't quite follow your argumentation. Why did you compile charcoal records over such a vast area? Are you expecting similar climatic trends

over the Holocene, over this entire area 1000s of km across? If yes, then you should make it clear why (monsoonal systems etc). If no, then it doesn't make much sense to compile all of these into one record. You should also discuss in more details the different temperature of production of charcoal and levoglucosan, what does it imply? What explains different fire temperature, and what fire would you then expect to explain your data.

A: We agree with your observation on the fact that an area of 1000 km could be too wide to be considered as reference for Paru Co. Due to the fact that also Anonymous Referee #2 commented on this topic, we decided to exclude this part of the work from the new version of the paper due to the fact that this confrontation with GCD was not improving the data interpretation.

Page 13: It would have been useful to show a record of changes in monsoon strength (e.g. precipitation) along your own records.

A: As suggested, we added dD per mill of C27 and C29 figure 4(a) and lithics% in figure 5(b), retrieved from Bird et al. (2014), which are used as Indian Summer Monsoon indicators. Figures 4 and 5 are now strongly different respect to the previous version of the paper.

Page 13: If you would rather not trust your PAH record as a fire record, then you should make it much clearer earlier on, and mention that you will then only discuss MAs. Given the high variability of your PAHs, it may be your best option.

A: Since we reanalyzed all the PAHs fractions, we found new interesting results and we were also able to calculate Ant/(Ant+Phe), IP/(IP+Bghi) and FluA/(FluA+Pyr) diagnostic ratios. Therefore, we improved the discussion and the comparison between PAHs and MAs (section 5, from P.10 of the new version of the paper).

Page 14 line 7: Are you now argumenting that the MAs peak at 5.6 is due to transport, and not fire activity? A bit earlier you were discussing the Bond Event 4 and monsoonal

precipitation. This is somewhat confusing. If both could play a role, you should add a summary sentences stating that.

A: In the new version of the paper we removed the association between MAs and Bond Event 4. On the contrary, we improved a lot the discussion of long-range transport, with a new paragraph called "5.4 Atmospheric transport" in which we tried to explain how one of the most probable source of levoglucosan seems to be atmospheric transportation from the South.

Smaller issues: Title: I recommend adding "Lake" to Paru Co

A: The new title is "Fire, vegetation and Holocene climate in a south-eastern Tibetan lake: a multi-biomarker reconstruction from Paru Co".

Abstract, line 24: I would briefly explain why PAHs decreases but MAs remain high. What is the distinction to the intense biomass burning during the early Holocene where both were high?

The new data show that high molecular weight PAHs peak in the early Holocene, and we address this fact to intense local fires and elevated burning temperatures. We explain this fact in the discussion section (P.11 L.1-2): "The high concentrations of higher molecular weight PAHs during the early Holocene could be explained with local fires of greater combustion temperatures, due to the fact that higher number of rings requires greater burning energy (Denis et al., 2012) [...]" .

Page 2, line 5: In the sentence just before you state that fire contribute to greenhouse gases, this contradicts the end of the sentence here.

A: We agree and revised accordingly, now the sentence sounds as: "The impacts of greenhouse gases and associated global climate change on the frequency, intensity, duration, and location of biomass burning are not well understood and the contribution of fire emissions to past and future atmospheric composition are also unclear (IPCC, 2014)".

Page 2, line 6: Why do you use "therefore"? I don't see a clear link into this last sentence. It makes sense that these sentences are in the introduction, but there is not much flow, or logical order to these sentences. They just seem to be put together. Please improve the argumentation.

A: We modified the sentences according to your suggestion. The new sentences sound as: "However, a recent study found that the synthesized Holocene fire record in eastern monsoonal China strictly tracks global atmospheric $CO_2$ concentration from Antarctica (Xue et al., 2018), but it is still not clarified which one between fire and $CO_2$ triggered the other. Therefore, more studies would be needed to improve human knowledge about past and present biomass burning events, which need to be characterized and accurately mapped in order to investigate interactions with weather, climate, and landscape dynamics over a range of spatiotemporal scales."

Page 2, line 13: (and throughout the manuscript) List references in chronological order, the oldest one first, the most recent one last)

A: Thanks for the indication, however, Climate of the Past does not require chronological order for the in-text citations, as it is explained in the website https://www.climate-of-the-past.net/for_authors/manuscript_preparation.html: "In terms of in-text citations, the order can be based on relevance, as well as chronological or alphabetical listing, depending on the author's preference". We have chosen the alphabetical order and we checked to be consistent throughout the manuscript.

Page 2 line 16: Can you specify what type of ecosystem processes?

A: We modified the end of the sentence, that now sounds as: "and other environmental processes such as vegetation growth, detrital influx, volcanic eruptions".

Page 2, line 18: Ice cores from where, also the TP?

A: In order to be more clear, we completely rewrote the sentence in this way: "Within the Tibetan Plateau (TP), only a few studies examine past biomass burning by using

charcoal (Herrmann et al., 2010; Miao et al., 2017) or black carbon. Polycyclic aromatic hydrocarbons (PAHs) are reported in the lake sediments from the Tibetan Plateau (TP) spanning the last 2 centuries (Yang et al., 2016). Monosaccharide anhydrides (MAs), ammonia and black carbon in ice cores have been used as combustion proxies and indicators of fire on or influencing the Tibetan Plateau, but these records only cover the last century (Kaspari et al., 2011; Ming et al., 2008; Shugui et al., 2003; Xu et al., 2009; You et al., 2016b)".

Page 2 line 26: I don't see what you mean by specific environmental conditions?

A: The specific environmental conditions are explained after in the text, concurrently with the more detailed description of every class of biomarkers.

Page 2 line 31: I would clarify here that the following list of marker you are discussing are Mas

A: We agree and revised the sentence in this way: "Within the listed biomarkers, MAs are specific tracers of vegetation combustion".

Page 4 line 25-26: Please rephrase this sentence: the first part of the sentence is about difficult access to paleoclimate archives and then you mention few investigations into species diversity and plant communities. Where is the relationship?

A: We rephrased the sentence, that now sounds as: "However, its remote nature re-strains access to possible paleoclimate studies, resulting in relatively few investigations of past species diversity and plant community changes".

Page 5 line 19: Define ecosystem functions or use another word, e.g. vegetation distribution?

A: We incorporated your change in the sentence as following: "More recently, human activities and related climate change have significantly altered the regional hydrology and vegetation distribution of the plateau, with degeneration of plants that led to deser-tification and frequent dust storms (Wang et al., 2008)".

Page 5 and Figure 1: It would have been more useful to have a more precise catchment map showing these features thant the large google map on figure 1 or the satellite picture showing only the lake.

A: In the new version of the paper we added a focus on atmospheric transport to the Tibetan Plateau. That is why in our opinion figure 1(a) is important to understand the continental position of the lake within the neighboring geographic areas. We have just zoomed in on the study region.

Page 6 Line 8: The 137Cs determination method is not cited here, whereas the radio-carbon is.

A: We added the expression "determined by direct gamma counting" in order to specify the 137Cs determination method.

Page 8 line 7: How did you obtain wet density?

A: We calculated wet density with this formula: dry density (g/cmˆ3) + (water content (%) * water density (g/cmˆ3)).

Page 8 line 11: for the other ratios you clearly state what they are useful for. You should do the same with the ACL, what can it tell you?

A: We briefly added the significance of ACL. The sentence now sounds as: "the average chain length (ACL), representing the composite of longer and shorter n-alkanes between the chain length range of 21 to 33 and indicating the prevailing length".

Page 8 line 29: please provide these latitude an longitude ranges. How many records did you compile in total?

A: Since we removed the comparison with the GCD, this information can be neglected.

Page 9 line 7: or you had erosion of older compounds in the catchment (soil).

A: Thanks for the suggestion. As already specified earlier in the responses, in the new

version of the paper we decided not to discuss the data of the deepest part of the core.

Page 9 line 30: Why didn't you look at the correlation between BiSi and PAHs and TOM and PAHs?

A: The correlation between PAHs and TOM was indicated at page 10 line 14 in the discussion paper. However, due to the fact that the new PAHs data are different, we did not consider in the discussion these correlations.

Page 9 line 33: It would have been good to summarise here how the link ISM-BiSi works.

A: Thanks for the tip. In the new version of the paper, BSi is not used as indicator for ISM. Instead, we used lithics(%), inserted in figure 5(b), because more intense rainfall result in greater lithic deposition (Bird et al., 2014).

Page 10 line 1-9: These correlations should be presented when you first describe similar trends of both PAHs and MAs, at the beginning of the section!

A: We agree with the comment and, in the new version of the paper, we present the correlation values at the beginning of the result section.

Page 10 line 2: the (negative) correlation between MAs and TOM was larger than this (-0.54) and with a more significant p-value. Maybe there is something to discuss there, even though it is not positive as you expected.

A: Thanks for the comment. The correlation at page 10 line 2 was between PAHs and MAs. The correlation between TOM and MAs was at page 9 line 29. r = -0.54 was referred to the correlation between MAs and BSi, not to MAs and TOM. For this reason your question is not totally clear to us. By the way, we were not expecting positive correlation between MAs and TOM, because it would have been signified a relationship of MAs with organic matter.

Page 10 line 5: how does it vary? Does it vary with time, depending on the main

climate?

A: That sentence was removed from the paper. However, an explanation about MAs catchment area is given in the new section 5.1 (paleofire activity) where we say that "MAs are capable of travelling hundreds of kilometres (Schüpbach et al., 2015; Zennaro et al., 2014)".

Page 10 lines 5-9: These last two sentences should probably be moved to the discussion section.

A: Thanks for the comment, we incorporated your changes in the new version of the paper. Now, section 4 is dedicated only to the mere description of the results.

Page 10 line 7: Unfortunately we don't see the original concentration in your figures, only the fluxes.

A: The new figures strongly changed due to the facts that we have some new data and that we decided to use concentrations instead of fluxes, because data discussion is focused between 1.347 and 10.768 cal ky BP, when sedimentation rate is constant.

Page 10 line 8: can you specify what you mean by "biogenic origin"? combustion of biomass is also biogenic for me

A: In the new version of the paper we the confrontation between PAHs and TOM was not considered for the data interpretation, so that sentence is no more present.

Page 10 Line 13-15: I don't fully understand this sentence, can you rephrase it to be more clear?

A: Due to the substantial modifications in the results, section 4 and 5 of the new version of the paper are completely changed and this sentence is no more present.

Page 10 line 15: Is the statistic done only for this interval, or for the entire core? please specify

A: Due to the substantial modifications in the results, section 4 and 5 of the new version of the paper are completely changed and this sentence is no more present.

Page 10 line 19: please provide some examples of these changes in the terrigenous environment.

A: Due to the substantial modifications in the results, section 4 and 5 of the new version of the paper are completely changed and this sentence is no more present.

Page 10 line 28: It would be useful to either state the published range in the text, or in the figure. Are those ranges for L/M, or for L/(M+G)

A: As you suggested, we incorporated a better description of the L/M ranges in the new version of the paper, in the new section 5.2 "Combustion sources", where we state, for example: "In addition to the PAH ratios, L/M ratios can also help determine combustion sources. L/M emission ratios ranging between 0.6–13.8 may be due to softwood combustion, while ratios between 3.3–22 depict hardwood burning, and ratios 2.0–33.3 may be due to burning grasses".

Page 11 line 1: the second part of this sentence is rather vague. n-alkanes do not record all organic input inot the lake, and they also record organic production within the lake.

A: We revised the sentence according to your suggestion: "Past vegetation changes can also be derived by variations in n-alkane ratios, as n-alkanes can record the organic input into and within the lake".

Page 11 line 15: Here you should list the other FeSts, especially those that would have indicated the presence of humans.

A: Due to the substantial modifications in the results, section 4 and 5 of the new version of the paper are completely changed and this sentence is no more present.

Page 11 Line 16-17: Are there other information (e.g. archeology) which could support

this finding? No known settlement in this area, too high elevation, ...?

A: Due to the substantial modifications in the results, section 4 and 5 of the new version of the paper are completely changed and this sentence is no more present.

Page 11 lines 19-23: All this first part should be in the method section.

A: Due to the substantial modifications in the results, section 4 and 5 of the new version of the paper are completely changed and this sentence is no more present.

Page 11 line 31: I only counted 3 colour bars where the arrows go in the same direction, that's pretty bad as a similarity...

A: Due to the substantial modifications in the results, section 4 and 5 of the new version of the paper are completely changed and this sentence is no more present.

Page 12 line 31: I wouldn't call this composite record regional, it's almost continental

A: Due to the substantial modifications in the results, section 4 and 5 of the new version of the paper are completely changed and this sentence is no more present.

Page 13 line 5: How do these different burning temperature occur? You need to discuss this point in further details. Would we then have low or high temperature fire during this interval, and why, what caused this type of fire?

A: Due to the substantial modifications in the results, section 4 and 5 of the new version of the paper are completely changed and you would see that references to different combustion temperatures are made frequently throughout the text. For example: "If we assume that low molecular weight PAHs degrade at 500 °C, we have to assume that MAs may also degrade at this temperature, as maximum concentrations occur at burning temperatures centred around 250 °C".

Page 13 line 14: The PAHs show a clear minimum at 8kyr. The Sum of MAs show a peak from 8 to 9 kyr BP. The 8.2 cal ky BP was a short and abrupt event, if the ISM was peaking then, I would not expect a 1000 year long dry interval.

A: Due to the substantial modifications in the results, section 4 and 5 of the new version of the paper are completely changed and this sentence is no more present.

Page 15 line 4: that warm period would fall right into the 8.1 to 7.2 cold intervals (1-2 degrees cooling) you are mentioning a few lines up...that's contradictory

A: We agree and revised the sentence in order to be more clear. The new sentence sounds as: "However, within this warm period, the climate had a sudden, intense change between 8.1 and 7.2 cal ky BP with temperatures 1-2 °C below early and mid-Holocene levels and forests retreating downslope."

Page 15 lines 6-7: please specify the time interval here. As you were just mentioning relatively young intervals (<2.7 kyr BP) where you don't have fire records it is confusing. Do you mean for the early Holocene?

A: Due to the substantial modifications in the results, section 4 and 5 of the new version of the paper are completely changed and this sentence is no more present.

Page 15 line 18-19: I wonder which one (forest or shrub) tend to have more fire? I also wonder if the presence of forest or shrub has an influence on the fire temperature?

In general, woody fires tend to have higher temperatures (350-550 °C) respect to grass fires (120-250 °C). Due to the fact that forest and shrubs are both woody plants, there is no much difference between them, but it depends mostly on the weather/climate. Fire temperatures are influenced by both quantity and quality of the fuel. In woody vegetation, backfires frequently burn longer and deeper but headfires are hotter.

Page 15 line 22-24: In this sentence is is hard to follow what the observations where, and what are the suppositions, could you reformulate more clearly what has been observed, and what is assumed?

A: We clearly reformulated the sentences, that now sound as: "After 5.2 cal ky BP, lake levels decreased, probably causing opposite fluctuations in both ACL and Paq, suggesting diminished ISM rainfall, reduced clastic deposition, and lowered lake levels,

which leading to an invasion of the littoral zone on the core site and an increase in sand deposition (Bird et al., 2014). The fluctuations in both ACL and Paq are consistent with these lake level changes (Figures 3 and 4)."

Page 15 line 32-33: I don't follow your argumentation. How can you infer that this mechanism would also occur in the sediments?

A: Due to the substantial modifications in the results, section 4 and 5 of the new version of the paper are completely changed and this sentence is no more present.

Page 16 line 1-2: leaf waxes can also be abraded and transported by the wind, as well as in streams in suspended sediments, leaves are not necessarily requested for their transport and deposition

A: Due to the substantial modifications in the results, section 4 and 5 of the new version of the paper are completely changed and this sentence is no more present.

Page 16 line 6: I don't understand what you insinuate here? What details would you look into? Please be more specific and/or provide examples.

A: Due to the substantial modifications in the results, section 4 and 5 of the new version of the paper are completely changed and this sentence is no more present.

Page 16 line 25: No, they can also be transported quite far by winds.

How much do you mean with "quite far"? In the new version of the paper we found that CPI and L/M have a slight positive correlation (r = 0.31, p-value = 0.03) suggesting that both local and regional sources are possible.

Page 17 line 1: what do you mean by paleoreconstruction information, this is too vague, could be paleotemperature, paleoprecipitation...

A: Due to the substantial modifications in the results, section 4 and 5 of the new version of the paper are completely changed and this sentence is no more present.

Page 17 line 27: You should not discuss something for the first time in the conclusion. This expansion of Bronze Age civilization should have been mentioned earlier. In the text, you mention 4.2k as the collapse of Chinese Neolithic cultures

A: Due to the substantial modifications in the results, also section 6 of the new version of the paper is completely changed and this sentence is no more present.

Page 31 – figure 5: Your lake is located at 30N, why do you use the insolation at 40N?

A: All the figures from 2 to 5 have strongly changed. Now we used insolation a 30° N in figure 4(e).

Page 31 – figure 5: There are dots in between the Paq and the ACL graphs, to which graph do they belong to?

A: All the figures from 2 to 5 have strongly changed. Paq and ACL are now shown in figure 4(b,c).

Page 31 – figure 5: I wonder what signal you would obtain if you were to use the same 5pt moving average on the Paq and the ACL. It seems to me that the Paq data without moving average show a signal similar to the sitostanol

A: All the figures from 2 to 5 have strongly changed. Paq and ACL are now shown in figure 4(b,c) and compared to lake level changes from Bird et al. (2014).

Page 31 – figure 5. I am not quite convinced by the insolation driven trend. Or at least I don't think this is the signal you should be looking for in your n-alkanes data over the Holocene. I am also not convinced by your green bars highlighting similar oscillations in Norm 31 and sitostanol. The youngest peak is relatively coeval, the one before is already almost opposite (sitostanol peak is closer to the Norm31 minimum than to its maximum). The third Norm31 peak is not coeval with any peak in Sitostanol. And the 4th peaks are again quite offset.

A: All the figures from 2 to 5 have strongly changed. The comparison between Norm31

and Sitostanol was removed, since it was not considered significant within the new data interpretation.

Page 32 Figure 6: The figure is of poor resolution. The blue dots on the map in panel a are not described/explained. If not used these should be removed. The oscillations in panel b (charcoal) can barely be seen. The curve and its envelope (dotted curves) are not explained. I suggest deleting "resulted" and "analysis" from the figure caption. The figure caption should also describe what the blue green and red arrows are for.

A: Figure 6 was removed, since the GCD comparison was no more informative within the new data interpretation.

Technical corrections Page 2, line 11: delete provided

A: We agree and revised accordingly, the new sentences sounds as: "Lake sediments archive high-resolution histories of sediment flux, as well as climatic, hydrological and ecological changes, as long as the lakes preserve sediments through time"

Pag 2 line 28: delete "in buried sediments"

A: We agree and revised accordingly, the new sentences sounds as: "Significant concentrations of these compounds are present in soil and sedimentary archives with ages older than 10 cal ky BP".

Page 2 line 31: You could add "and longer timescales" after "the Holocene"

A: We agree and revised accordingly, the new sentences sounds as: "suggesting that degradation, if happening, is a low-kinetic process (Battistel et al., 2016) and that these compounds resist over the Holocene and longer timescales"

Page 3 line 17: I would here mention "diverse distribution of chain length"

A: We agree and revised accordingly, the new sentences sounds as: "Different types of plants have diverse distribution of n-alkanes chain-lengths".

Page 4 line 1: replace "anthropological" with "archeological"?

A: With "anthropological" we mean a wider evidence of human presence (FeSts, human-related pollens, . . .) respect to "archeological", that is only related to archeological findings.

Pag 4 line 2: replace "quantification" with "determination" (I wouldn't say that we can truly quantify the presence of humans. We are not there yet.

A: We agree and revised accordingly, the new sentences sounds as: "Revealing human presence in lake catchments often relies on anthropological evidence, but advances in proxy development during the past two decades now allows determining of the presence of humans or pastoralism through steroid fecal biomarker concentrations".

Page 5 line 5: You could mention the Younger Dryas & Bolling Alerod in your text.

A: We agree and we added "in the context of Bølling–Allerød and Younger Dryas events in the region" to the sentence.

Page 5 line 12: You could maybe use "superimposed on these oscillations" instead of "even with these oscillations".

A: We agree and revised accordingly, the new sentences sounds as: "Superimposed on these oscillations, the general temperature trends affecting the TP include warm and humid climate in the early to mid-Holocene, as registered in sediments and dust deposits".

Page 8 line 3: I would rather use data "analysis" instead of "elaboration"

A: We agree and revised accordingly. The title of the section is now "3.4 Data Analysis"

Page 8 line 9: Replace "significative" with "significant"

A: We agree and revised accordingly, the new sentence sounds as: "In order to help data interpretation, 2-tailed Pearson's correlations were calculated in R with a 95%

confidence interval (Supplement S3) with statistically significant results when p-values are < 0.05".

Page 8 line 11: what do you mean with "useful for work"? I would rather say "useful for our study, or for our interpretation"

A: We agree and revised accordingly, the new sentence sounds as: "N-alkanes ratios useful for our study include [...]".

Page 8 line 30: Here you should refer to the figure presenting this data

A: Due to the substantial modification of the paper, this part is no more included in the new version.

Page 9 line 17: use "up to" instead of "touching values till"

A: Due to the substantial modification of the paper, this part is no more included in the new version.

Page 9 line 21: remove the comma

A: Due to the substantial modification of the paper, this part is no more included in the new version.

Page 13 line 27: delete the second "not"

A: Due to the substantial modification of the paper, this part is no more included in the new version.

Page 15 line 2: Is there a word missing? ("a limited abrupt to"???)

A: Due to the substantial modification of the paper, this part is no more included in the new version.

Page 15 line 13: I would rather use another verb, for instance "place the Paru Co..."

A: Due to the substantial modification of the paper, this part is no more included in the

new version.

Page 15 line 15: specify the time interval considered

A: Due to the substantial modification of the paper, this part is no more included in the new version.

Page 15 line 26: I would rather call this "long term trend" than "millenial scale".

A: Due to the substantial modification of the paper, this part is no more included in the new version.

Page 15 line 29: can you indicate by how much on average?

A: Due to the substantial modification of the paper, this part is no more included in the new version.

Page 15 line 30: same here, by how much? is it significant?

A: Due to the substantial modification of the paper, this part is no more included in the new version.

Page 16 line 4: "it seems that"

A: Due to the substantial modification of the paper, this part is no more included in the new version.

Page 16 line 11: slash and burn -> at which time?

A: We added "over the past 4600 years (Miehe et al., 2006)" in the sentence.

Page 16 line 11: Please indicate Lhasa on Figure 1.

A: We agree and revised accordingly.

Page 16 line 12: where exactly? Please show on the map.

A: We agree and revised accordingly.

Page 16 line 16: compared to (or "with respect to")

A: Due to the substantial modification of the paper, this sentence is no more included in the new version.

Page 16 line 32: Comparing data that resulted from (or originated from)

A: Due to the substantial modification of the paper, this sentence is no more included in the new version.

Page 16 line 33:"processes that happened"

A: Due to the substantial modification of the paper, this sentence is no more included in the new version.

Page 19 line 6: The journal and pages are missing.

A: We checked and added the correct missing details.

Page 29: Specify in the figure caption that the BSi axis is inversed

A: Thanks for the observation, but now BSi in no more used in our graphs.

---

## Author Comment (AC2) · 9 Jul 2018

PAGES Data Review Team The PAGES Data Stewardship Integrative Activity seeks to advance best practices for sharing the data generated and assembled as part of all PAGES-related activities. The CP Special Issue, "PAGES Young Scientists Meeting 2017" is part of this PAGES activity. The co-editors of the Special Issue are reviewing the data availability within each of the CP-Discussion papers in relation to the CP data policy (https://www.climate-of-thepast. net/about/data_policy.html) and current best practices. The editor team is making recommendations for each paper, with the goal of achieving a high and consistent level of data stewardship across the Special Issue. We recognize that an additional effort will likely be required to meet the high level of data stewardship envisaged, and we appreciate the dedication and contribution of the authors. This includes the use of Data Citations (see example below). Authors are also strongly encouraged to deposit significant code into a suitable repository and to cite it using a Data Citation. We ask authors to respond to our comments as part of the regular open interactive discussion. If you have any questions about PAGES Data Stewardship principles, please contact any of us directly. Best wishes for the success of your paper. YSM Special Issue editor team (E. Dearing Crampton-Flood, D.S. Kaufman, R. Barnett, M.F. Loutre, M.N. Evans, S.C. Fritz, C. Tabor, Y. Zhang, E. Razanatsoa, and H. Plumpton) For this paper: All papers submitted to Climate of the Past must include a Data Availability section that details the location of the data that were used as input to the study, including previously published data that were used for comparison purposes, and the data that were generated by the study.

(1) Research input data- The paper makes use of the Global Charcoal Database (GCD). In order to adhere to the Data Policy for Climate of the Past, persistent identifiers (doi or URL from NOAA Paleoclimatology), or full data citations to the primary data must be included in the Data Availability section.

A: We thank the PAGES Data Review Team for the comments and suggestions.

Fig. 1 (c) includes data from the TRMM dataset, which is not in the reference section. Please add the appropriate data citation to the paper.

A: We have added the appropriate data citation in figure 1(c).

Fig. 3 includes proxy data from Bird et al. (2014). We were delighted to discover that the data are already available in a public repository at https://www.ncdc.noaa.gov/paleo/study/16399. Add a data citation for this dataset to the figure caption and to the Data Availability section.

A: Now figure 3 is different and it includes data from Zhao et

[Figure]

al. (2011) that we appropriately cited, adding the data citation as well:http://apps.neotomadb.org/Explorer/?datasetid=14619. Data from Bird et al. (2014) are now included in the figures 4(a,d) and 5(b). We inserted the appropriate data citation to the figures' captions and to the Data Availability section.

Fig. 4 (c) includes a summary of pollen data from Tang et al. (2000) that was digitized from the original publication. Digitizing data from previous publications is a legitimate practice, but has important disadvantages. The accuracy is often degraded by the digitizing, and without essential metadata, the data are not available for reuse. Instead of digitizing previously published data, we encourage authors of this special issue to serve as data stewards by working with data generators to rescue and properly curate important datasets that are used in their paper. Please contact Tang to explain the reuse of the data and to offer to facilitate the transfer of the dataset to a data repository. Once deposited in a repository, the data can be cited with a data citation.

A: We agree with the fact that digitazing can reduce accuracy. However, we were not able to came in contact with Tang. We therefore decided to remove the comparison with Tang's dataset from figure 4 and we choose to compare our data with pollen dataset from Zhao et al. (2011), which is uploaded in Neotoma Data Publisher public repository (http://apps.neotomadb.org/Explorer/?datasetid=14619), displaying this data in figure 3(c).

Fig. 6. Include a data citation to the GCD in this figure caption and in the Data Availability section.

A: Due to the revision process we changed big part of the paper, dismissing the use of the GCD and therefore removing figure 6.

(2) Research output data – Biomarker data plotted in Figs 4a-b, 5b-e, 6c: Monosaccharide anhydrides (MAs), polycyclic aromatic hydrocarbons (PAHs), n-alkanes, fecal sterols and stanols (FeSts) – This paper presents new and valuable biomarker data for the south-eastern Tibetan Plateau during the Holocene. These new data must be

uploaded to a long-standing online data repository, and a data citation or URL link from NOAA Paleoclimatology for access to these data must be provided in the Data Availability section of the paper. Charcoal index plotted in Fig. 6b: Composite of previously published data from 43 sites – This is an important new summary of the regional fire history based on available charcoal records. The outcome of the synthesis (the time series with uncertainties) should be transferred to a repository along with a table of metadata that includes the name and location and reference for each of the 43 sites. This product could be included on the same landing page, with the same doi/NOAA URL along with the biomarker data.

A: All the research output data, except for charcoal index that was removed from the paper, are now available at this link: https://www.ncdc.noaa.gov/paleo-search/study/24410.

---

## Author Comment (AC3) · 9 Jul 2018

Anonymous Referee #2 General Comments: In this paper Callergaro et al. present results from a biomarker multiproxy reconstruction of fire and vegetation from lake sediments Holocene on the Tibetan Plateau. The methodology used in this paper and the scientific aims of this study are will within the scope of this journal. This paper applies a clever approach where multiple lines of environmental evidence (i.e. fire, vegetation, human/animal habitation) can be reconstructed from the same samples using a relatively streamlined workflow. Additionally, I appreciate the authors' tactic of using data from the GCD to interpret their fire data within a regional framework. Despite these

strengths, this paper could be improved by better presentation (figures) and clearer interpretation of the data. I found myself unable to follow the logic at times, and occasionally, the data and interpretations seemed at odds with each other. Adding more complete explanations of proxy interpretations (in both the text and the figures) may clear up some of this confusion. There were also times in the discussion where evidence from other studies was presented without being linked to the new data, and the new conclusions felt buried. Make sure to emphasize the novel contributions of your work and what it adds to the literature framework. I have outlined some more specific issues below. Making these improvements will greatly increase the readability of this paper and strengthen the arguments.

A: Thanks for your helpful review and your observations that are surely pivotal for improving our paper. We substantially revised and rewrote the paper following your indications. We have to say that we had some difficulties in replying to some of your observations, especially in the "technical corrections" part, due to the fact that the pages/lines that you indicated in the comments were not corresponding to the ones in the discussion paper file. We tried our best in finding the precise arguments in the paper and in responding to your questions. We reply to specific comments below.

Specific Comments: Page 8 Line 31: Are you using %BSi as a proxy for monsoon intensity? If so an added sentence explaining why would be helpful. Also did you measure %BSi or %TOM or is it from Bird et al 2014? Please specify.

A: We incorporated your suggestion in the new version of the paper where, however, BSi is no more used as indicator for ISM. Instead, we used lithics(%), whose data are inserted in figure 5(b). We also specified in the text that "more intense rainfall result in greater lithic deposition (Bird et al., 2014)". Moreover, we added dD per mill of C27 and C29, in figure 4(a), which are also used as Indian Summer Monsoon indicators. Figures 4 and 5 are now strongly different respect to the previous version of the paper. Finally, we specified, when necessary, that these data come from Bird et al. (2014).

Page 9 Lines 13: I don't know if concurrent increases between PAHs and TOM implies a specifically biogenic origin for PAHs, just that the total organics in the lake and the PAHs may have a similar source. Especially given that your aquatic/terrestrial indicators show an increase in terrestrial n-alkanes to the lake after 8 cal ky. Perhaps there is more windblown terrestrial material being added to the organic pool and that's why it's increasing? You touch on this in the next paragraph.

A: We agree with your observations. In the new version of the paper we dismissed the focus on PAHs and TOM, since, as suggested by Anonymous Referee #3, we reanalyzed all the samples for PAHs, including new target ions. The new obtained results are now shown in figure 2, that strongly changed. With the new PAHs data we calculated some diagnostic ratios such as Ant/(Ant+Phe), IP/(IP+Bghi) and FluA/(FluA+Pyr). Therefore, we improved the discussion and the comparison between PAHs and MAs, but the confrontation with TOM was considered to be pointless for the new paper. Moreover, we compared the aquatic/terrestrial indicators to the lake levels (from Bird et al., 2014) in a new figure 4(b,c,d).

Page 9 line 20: MAs are more water-soluble than PAHs, so this argument doesn't make a lot of sense

A: We agree with this point, indeed PAHs are more lipophilic than MAs. We talked about these differences in a new paragraph, that sounds as: "The explanation for the lack of levoglucosan and other MAs peaks during the period of the highest concentrations of PAHs (6.5-3 cal ky BP) may be due to: i) different burning temperatures and conditions, i.e. MAs are produced in smouldering and low temperature fires while flaming high temperature fires produce PAHs (Simoneit, 2002); ii) the lipophilic properties of PAHs, which have a low solubility in water (Haritash and Kaushik, 2009) while levoglucosan has a relatively higher water solubility, with an estimated half-life time of 5-8 days due to possible degradation from aquatic microorganisms who utilize the "free" form of levoglucosan (Norwood et al., 2013)".

Page 9 Lines 31-33: You need to explain and cite how you are interpreting these ratios and what is the difference between L/M and L/(M+G).

A: We incorporated your changes into the paragraph as the following: "Although MAs ratios cannot precisely point to the type of past burnt vegetation, these ratios can classify general vegetation types (Fabbri et al., 2009). However, due to the fact that galactosan presents a different biodegradation behaviour, the application of L/(M+G) ratio may be inadequate (Kirchgeorg, 2015). For this reason, we limited the discussion only to L/M ratio results. [...] In addition to the PAH ratios, L/M ratios can also help determine combustion sources. L/M emission ratios ranging between 0.6–13.8 may be due to softwood combustion, while ratios between 3.3–22 depict hardwood burning, and ratios 2.0–33.3 may be due to burning grasses (Fabbri et al., 2009 and references therein)".

Page 10 Paragraph on Line 5: ACL and Paq represent indices for differential terrestrial/aquatic inputs I'm not sure how that directly relates to interpretations of fire and vegetation change. Make it clear if you are relating this to lake levels and climate because these proxies don't explicitly address changes in terrestrial vegetation community.

A: Thanks for the request for clarification. We are using ACL and Paq in order to retrieve information on past vegetation changes in the lake catchment, since n-alkanes can record the organic input into and within the lake. We therefore clarified this point comparing these 2 ratios with lake level changes and climatic variations in the new figure 4 (a,b,c,d,e) where we parallel dD per mill of C27 and C29, ACL, Paq, lake level changes, summer insolation at 30° N.

Page 10 Line 19: Are you using this to say something about source area changes or vegetation community changes? This distinction is not clear.

A: If you refer to PAHs, in the new paper we discussed about their pyrogenic sources and no more about terrigenous sources (page 10 line 19). If you refer to sitostanol

(page 11 lines 12-13), due to the fact that human-related FeSts were below method detection limits, we suppose that its source have to be vegetation related.

Page 10 Line 23: Is there any evidence that sitostanol is correlated with grassy tissues? Citation?

A: We did not found these kind of correlation. However, derivation from vascular plants is reported (Vane et al., 2010).

Page 11 Lines 13-32: This paragraph is under "paleofire activity" but doesn't mention fire at all, just gives some climate context. Perhaps having a climate section would be useful?

A: We agree with your observation. The paragraph to which you refer is actually on page 12, lines 3-24. Indeed, we removed this paragraph from the section 5.1 and we put it in a new section: "5.4 Atmospheric transport".

Page 12 Lines 9-15: Throughout the manuscript, potential reasons for the differences between the MA and PAH records are mentioned, but no evidence is presented to support any of these interpretations over the other. Are there sedimentary changes (i.e. grain size) when these records diverge that indicate changes in transport to the lake? Are there changes in PAH ratios throughout the core that might indicate changes in transport/fire temperature? Perhaps there are inherent differences in the transport of MAs and PAHs due to their size/solubility differences? Why might the charcoal and PAH record correspond better than the charcoal and MA records? If these questions are explicitly addressed it will greatly strengthen interpretations of fire history. If you haven't already, I'd suggest reading Denis et al. 2012 Organic Geochemistry, which has a good discussion considering transportation/degradation/fire temperature differences in lake fire proxy records.

A: Thanks for the questions. We have tried to address all of them in the new discussion sections. Due to the fact that new PAHs data were obtained, a new interpretation is evidenced, explaining the divergence between PAHs and MAs as fire indicators. We also read Denis et al and used it for our interpretation. Are there sedimentary changes (i.e. grain size) when these records diverge that indicate changes in transport to the lake? We associated PAHs and lithics(%), evidencing that in some cases increased PAHs signal is concurrent with less lithics (drier periods). MAs, instead, are higher during the intense ISM of the early Holocene. Our explanation of this fact in the new paper sounds as: "Although PAHs are more of a local fire indicator than levoglucosan concentrations, PAHs are also affected by changes in atmospheric transport and associated precipitation. PAHs peak during periods of less intense ISM precipitation, as indicated by Paru Co lithics % in the periods 10.5-10.1, 7-5.8, 5.2-3.2 cal ky B (Figure 4). During these drier phases aridity could have increased regional fire activity (Section 5.1). However, this relationship between aridity and fire is not constant for the late Holocene Paru Co record as the increasing PAHs signal from 3 to 1.3 cal ky BP coincides with increasing lithic abundances that may be related to more ISM precipitation. Therefore, the 3-1.3 increasing PAHs could be related to a fire signal transported by ISM precipitation. Rainfall occurring together with or soon after with fire events scavenges PAHs particles from the atmosphere and increases deposition (Denis et al., 2012)". Are there changes in PAH ratios throughout the core that might indicate changes in transport/fire temperature? We calculated the ratios Ant/(Ant+Phe), IP/(IP+Bghi) and FluA/(FluA+Pyr) and we associated an absolute error to their values, due to the error propagation in the calculations. We plot these values in the new figure 2, in order to highlight that the use of PAH diagnostic ratios could be reconsidered due to high overlapping values and error propagation that may hinder the correct allocation of the source. Perhaps there are inherent differences in the transport of MAs and PAHs due to their size/solubility differences? Of course MAs and PAHs do not always record the same fire events due also to their size and solubility differences. PAHs, especially the heaviest ones, can record only local signal, whereas MAs can be both local and regional. In the new version of the paper, section 5.1 (paleofire activity) we reported that: "Higher molecular weight PAHs are more stable compounds compared to 3-4 rings PAHs. If we assume

that low molecular weight PAHs degrade at 500 °C, we have to assume that MAs may also degrade at this temperature, as maximum concentrations occur at burning temperatures centred around 250 °C (Zennaro et al., 2015 and references therein). In the Paru Co record levoglucosan concentrations are higher than PAHs during the early Holocene. Therefore, in order to explain this discrepancy, regional early Holocene fires must have been more frequent than local fires, producing high amounts of MAs, without excluding that atmospheric transport of levoglucosan to Paru Co was more efficient during the early Holocene. Therefore, this high abundance of levoglucosan may also be related to a regional signal, as MAs are capable of travelling hundreds of kilometres (Schüpbach et al., 2015; Zennaro et al., 2014)" and later, in section 5.4 (atmospheric transport) we said that "This monsoonal history may affect the transport of fire products to Paru Co. The difference between the Paru Co MAs and PAHs records may be influenced not only by the burning temperatures that produce the different products, as previously mentioned, but may also reflect changing atmospheric transport. MAs peak during the ISM maximum at Paru Co between 10 and 7 cal ky BP, which may reflect the long range transport of these fire aerosols associated with biomass burning on regional scales (Figure 4). MAs are generally considered as regional signals due to their ability to be transported longer distances than the more local PAHs, where this early Holocene levoglucosan peak may reflect either increased fire activity and/or changes in atmospheric transport. We may hypothesise that high levoglucosan concentrations during the early Holocene in Paru Co reflect the interplay between increasing influence of the ISM in the early Holocene resulting in wetter conditions and increase biomass on the southern TP (An et al., 2012) as well as increased Early Holocene winter monsoons causing a cold and dry climate on the north-eastern TP that is cited as a main driver for fire activity during this time period (Miao et al., 2017). Major transport to Paru Co could have come from the south via the ISM but, to best of our knowledge, no studies encompassing Holocene fire history exist from the possible southern source areas". Why might the charcoal and PAH record correspond better than the charcoal and MA records? We agree with your later comment on the fact that the comparison

with the GCD may no hold due to the too much wide continental area from which the sites were selected. Therefore, we decided to exclude this part of the work from the new version of the paper, due to the fact that this confrontation with the GCD was not helping with the data interpretation. Would be really interesting to compare our data with the GCD if this database would be richer in sites from the Tibetan Plateau and/or from the Indo-Gangetic Plain, Bay of Bengal and South Asia.

Page 12 Line 26-28: What particular climate effect would Bond events have on fire in Tibet? Link through a mechanism

A: Due to the substantial modifications in the results, thanks to the new analysis, section 4 and 5 of the new version of the paper are completely changed and the references to Bond events were considered to be pointless and removed.

Page 13 Lines 1-4: Like I said in the results, I don't think you can assume that the PAHs are biogenic. Not without some additional evidence. Have you tried to use some PAH degradation ratios? The two papers you cite here are good resources for these tools. I'd be interested to see if applying the appropriate ratios to your data provide evidence for degradation.

A: As we already said, due to the new data and interpretation, we did not assumed the PAHs are biogenic. We also calculated Ant/(Ant+Phe), IP/(IP+Bghi) and FluA/(FluA+Pyr) diagnostic ratios, that helped in the interpretation. A new paragraph describing the ratios sounds as: "PAH diagnostic ratios used in this study are Ant/(Ant+Phe), IP/(IP+Bghi) and FluA/(FluA+Pyr). Ant/(Ant+Phe) values generally discriminate between petroleum (< 0.10) and combustion (> 0.10) sources; IP/(IP+Bghi) distinguishes between different combustion sources, with values > 0.5 for grass, wood or coal combustion, values between 0.2 and 0.5 for liquid fossil fuel combustion and values < 0.2 for petroleum sources; FluA/(FluA+Pyr) is used to define the transition point (0.5) between petroleum and combustion ( Denis et al., 2012; Yunker et al., 2002a; Yunker et al., 2002b; Yunker et al., 2015; Zakir Hossain et al., 2013). In Paru Co

these ratios are plotted with absolute error bars (Fig. 2g, 2h, 2i), in order to highlight that the influence of error propagation from the original analysis to the ratio values should be carefully investigated when assigning sources from the ratios. Considering the error bars, the three ratios shows values > 0.10 for Ant/(Ant+Phe), > 0.5 for IP/(IP+Bghi) and > 0.5 for FluA/(FluA+Pyr) for the majority of the analysed samples". As you suggested, we also calculated LPAH/HPAH as index of degradation (Stogiannidis and Laane, 2015), obtaining all values >1, that would signify no degradation and petrogenic sources. However, the cited paper is more focused on urban areas, so we think that applying this to our Holocene samples may be pointless, since the resulted values >1 do not make sense in our interpretation as pyrogenic PAHs.

Page 13 Line 5-15: I appreciate bringing in this discussion of transport: : :but I feel like how this relates specifically to your data gets lost. Does the peak at 5.6 represent increased wind, fire or both?

A: Due to the substantial modifications in the results, thanks to the new analysis, section 4 and 5 of the new version of the paper are completely changed and this reference to 5.6 wind event was removed. In the new version of the paper we talk about atmospheric transport from a regional point of view, listing the air masses movements that possibly brought fire signal to Paru Co.

Page 13 Lines 14-15: Are there dust layers recorded at Paru Co?

A: Thanks for the question. There are no dust layers recorded at Paru Co. We removed this paragraph from the new version of the paper.

Page 14 Lines 4-13: The times you highlight as having higher vegetation density are not all correlated with your fire records. Make sure you state when this mechanism applies and when it does not, and why that may be.

A: Thanks for the comment. We incorporated your suggestion in the new version of the paper, checking to be consistent and not to contradict the interpretation of the different

proxies. For example: "PAHs values are low in the early Holocene where, instead, tree pollen values are quite high. However, in the mid-Holocene PAHs contain higher concentrations from 6.5 cal ky BP, concurrent with a peak in the percentage of tree pollen. The subsequent decreasing trend in tree pollen, from 4.7 cal ky BP onward, is associated with a drying and cooling climate that may have intensified fire as recorded by PAHs in Paru Co, creating a positive feedback resulting in even more decreasing tree coverage. This decreasing trend in tree pollen reaches its lowest values after 3 cal ky BP. The regional wetter climatic conditions during the early and mid-Holocene may have favoured forest expansion, where this biomass became available for successive burning during the more arid climate of the late Holocene, when PAHs show indeed an increasing trend".

Page 15: I think that you need to address explicitly how the vegetation changes (i.e. shrubs versus trees) that are observed in the record are (or aren't) tied to your record of fire activity.

A: We agree and revised the paper according to your suggestion. Section 5.2 (combustion sources) and 5.3 (vegetation in the lake catchment) are now more focused on the relationship between fire and vegetation. For example, in the new version of the paper, we describe L/M comparing it to CPI: "In order to obtain more information from the burning conditions, we compared CPI values to L/M and PAHs. Considering that PAHs and n-alkanes are both local indicators, variations in CPI corresponding to spikes in local fire markers may link combustion and vegetation types demonstrated by n-alkane abundances. While no correlation exists between PAHs and CPI, the CPI and L/M have a slight positive correlation (r = 0.31, p-value = 0.03). Medeiros and Simoneit (2008) found that the n-alkanes in green vegetation smoke contained distributions ranging from C23 to C35, with strong odd-to-even carbon number predominance evidenced by CPI ranging from 9 to 58. MAs are better at recording smouldering fires than are PAHs, which may in part explain the similarity between MA and CPI variability through time. The Paru Co CPI values peak around 10 cal ky BP, in the period between 7.8 and 3.5

cal ky BP, and at 2.3 cal ky BP, with values up to 41.2, similar to the peak distributions of L/M. Another argument for relationship between CPI and MAs fire is the fact that lower temperature fires (MAs) essentially steam-distil the vascular plant lipids into the smoke, while high-temperature fires (PAHs) can result in decrease of the CPI[MOU1] (Schefuss et al., 2003 and references therein). In addition, the distance from the vegetation to the sediments may influence the CPI record as plants that are in or near the water pools contain shorter carbon chains, whereas more distant plants have higher CPI values (García-Alix et al., 2017). From these considerations can be assumed that, when CPI parallels L/M, fire from the surrounding areas, and not only near the lake catchment, could have been recorded".

Page 15 Lines 2-6: Leaves deposited into the lake are not the only source of terrestrial alkanes, they can also be ablated off leaves and transported by wind.

A: Due to the substantial modifications in the results, thanks to the new analysis, section 4 and 5 of the new version of the paper are completely changed and this part was considered to be pointless and removed.

Figure 3. Instead of color boxes interpreting the ISM instead show the original proxy record you are using for those interpretations. The Dalkane record from this same core from Bird et al. 2014 would be perfect to show plotted against proxies that you argue are influenced by ISM variations.

A: Thanks for the suggestion, we incorporated your changes in the new figures, which all strongly changed, where figure 2 compares PAHs, MAS, and PAHs ratios; figure 3 shows L/M, CPI, tree pollen, and PAHs; figure 4 shows Dalkane, ACL, Paq, lake levels, and insolation; figure 5 shows PAHs, lithics, MAs.

Figure 4. A and B could use some interpretive annotations. What to high/low MA ratios mean in terms of vegetation community? This is something that is unclear throughout the entire manuscript. How are these ratios interpreted and why do they differ? Adding some interpretive lines (like in 4C) would be very helpful. 4C. What is meant by arbitrary

units? Explain this in the caption.

A: Thanks for the suggestion, however it would be very confusionary to put interpretative annotations near to L/M plot, due to the intense overlap between ratio values. Moreover, we dismiss the use of Tang's data, preferring the dataset from Zhao et al., 2011 (http://apps.neotomadb.org/Explorer/?datasetid=14619)

Figure 5. I think plotting these records with a moving average obscures some important variability. It almost seems that the ACL and Paq have millennial cycles. Looking back at Bird et al. 2014, this seems to vary with reconstructed lake levels, rather than insolation as you argue. I think plotting that data in the same figure as these would really strengthen the interpretability of these proxies. Also, ACL and Paq and Norm31 and Sitostanol are really proxies for different things (aquatic v. terrestrial vegetation and changes in terrestrial community). It would be much clearer to separate these proxies into different figures. Perhaps, one figure with lake levels, insolation, ACL and Paq, and then add Norm31 and Sitostanol to Figure 4, which is your terrestrial vegetation figure.

A: We agree with your comment and we created the new figures according to your suggestion, except for Norm31, which was reputed not significant within the new data discussion.

Figure 6: Pg. 10 line 33 it is stated that the charcoal data from the GCD was drawn from a 1000 km radius from the Paru Co site: : :However, after a quick check on Google maps, I found all of the red dots on the Fig 6a map are actually more that 1000 km away from the site. Please address this contradiction and correct how the GCD sites were chosen in the text. Additionally, this large area of integrated charcoal records is potentially problematic because this data is drawn from a large continental area and the assumption that these sites are subject to the same climate conditions as the Paru Co site may no longer hold.

A: As already stated, the comparison with the GCD was removed, and consequently also figure 6 was deleted.

Figure 6 B and C: The shaded regions and arrows are not explained in the caption. And if they are showing correspondence then it almost seems that these records hardly reflect each other (which is discussed a bit in the text). Why is the PAH record not shown? I know it is highly variable, but it actually may correspond better to the charcoal record: : :if this is the case, then what are the implications for your data?

A: As already stated, the comparison with the GCD was removed, and consequently also figure 6 was deleted.

Technical Corrections:

A: We are sorry in saying that it was in some cases impossible to find where to address the requested changes, since we think that the indicated page/line references are wrong. We indicated this cases with "n.a."

Page 2 Line 1: The dependent clause of this sentence is unclear

A: n.a.

Page 2 line 13-15: perhaps an i.e. style list of just three or so methods

A: n.a.

Page 2 Line : move "in the last Century" to after "biomass burning"

A: We are not sure if you refer to "biomass burning" of page 2 line 1 or page 2 line 5.

Page 4 Line 6: Perhaps this would be a good place to introduce Neolithic/bronze age societies you talk about in the discussion.

A: n.a.

Page 4 Line 13: cite Figure 1a

A: n.a.

Page 5 Line 30: cite Figure 1d

A: n.a.

Page 6 Line 18: new paragraph at "Each Sample"

A: We revised accordingly. (found at page 6 line 26)

Page 7 Lines 18-19: What company did you obtain your standards from?

A: We added the companies and the sentence now sounds as: "We created response factors containing all of the target compounds as well as internal standards molecules (13C labelled acenaphthylene, phenanthrene and benzo[a]pyrene - Cambridge Isotope Laboratories, Inc; hexatriacontane and cholesterol-3,4-13C2 - Sigma Aldrich)".

Page 8 Line 11-12: This topic sentence does not fit the content of the paragraph. Additionally, I think you actually end up arguing that these records don't agree?

A: We suppose you refer to MAs and PAHs initial interpretation (page 9 lines 22-23). Section 4 was completely rewritten and the sentence was removed.

Page 9 Lines 10-13: This sentence is confusing and potentially unnecessary since you elaborate on it in the results.

A: n.a.

Page 10-11 Section 4.3: This reads like a methods section rather than results. Move this to methods and instead describe the trends you see in your analysis in the results.

A: We revised the paper removing section 4.3 (GCD results).

Page 11 Line 29: delete "a" before decreasing

A: The sentence was completely rewritten in this way: "The cooling trend after the Holocene Climatic Optimum (6.5-4.7 cal ky BP) correlates with decreasing solar inso-lation (Zhao et al., 2011), resulting in the decreasing strength of the Asian monsoon systems and in a drier climate across much of the TP".

Page 12 Lines 15-19: This is a rambling sentence; perhaps splitting it into two would

make it clearer?

A: n.a.

Page 15 Line 6: you seem to be missing a word after sedimentological

A: The sentence containing "sedimentological" was removed due to the intense rewriting process.

Page 15 Line 8: delete "seems that and add "also" between "could" and "help"

A: The sentence to which you refer was removed due to the intense rewriting process.

Page 15 Line 26: I'm confused what you mean with "except when associated to PAHs"

A: The sentence to which you refer was removed due to the intense rewriting process. By the way, we meant that when MAs parallel PAHs the most probable interpretation would be local fires; instead, when these 2 fire proxies show different behaviour, the MAs signal could be related to regional/continental fires.

Page 15 Line 28: Citation needed Bush and McInerney (2013) GCA and/or Diefendorf et al (2011) GCA.

A: The sentence to which you refer was removed due to the intense rewriting process.

Page 15 Line 27: Be consistent with the use of pollens or pollen, don't switch off

A: We revised the paper in order to be consistent with the use of pollen data / pollen records / pollens.

Page 16 Line 28: I think this is the first time you mention Bronze Age civilizations. You should elaborate on this earlier in the paper.

A: The sentence to which you refer was removed due to the intense rewriting process.

Page 16 Line 32: This is an abrupt way to end. Perhaps add a sentence of significance or implications?

A: The sentence of significance/implication were put at the beginning of the conclusions, which, in general, have been strongly modified due to the major revision process.

Figure 1. Include what the dates in 1D are based on (14C?).

A: We revised the caption incorporating your suggestion, as following: "(d) Plot of the age/depth model for Paru Co according to Bird et al. (2014) based on AMS 14C"

---

## Author Comment (AC4) · 9 Jul 2018

Anonymous Referee #3 General Comments: In this manuscript, Callergaro et al. demonstrate the usefulness of a multi-proxy approach to reconstruct fire and vegetation change throughout the Holocene from a lake sediment record in the Tibetan Plateau. The research objectives and methodology used in this study are within this journal's scope. This manuscript attempts to reconstruct fire, vegetation change, and human presence nearby the lake using biomarkers, and discuss how the results and other regional analyses of fire and climate compare. This manuscript presents a unique and novel record for the TP region. However, this paper could be improved with better interpretation of data, as well as better figures and presentation of data. The data and the author's interpretation seemed to conflict with eachother, particularly with the fire records presented. Adding more background information, discussion, and analysis of some of the records (detailed below) could drastically improve this paper and the discussion of results. I have presented some more specific issues below, listed from "Major Comments" to "Minor Comments" to "Figure Comments". Making these improvements will greatly increase the readability of this paper and strengthen the arguments.

A: We thank you for your kind revision and useful suggestions that helped us improving data interpretation, figures and the whole paper. We reply to the specific comments below.

Major Comments: I'm not sure I follow the PAH argument. In the manuscript, you argue that PAH track local and regional fire activity, along with MAs, in the early portion of the record (10.7- 8.7), but may switch to having biogenic origins after 8.7, just due to the fact that they correlate slightly with TOM. Looking at Fig 3, it seems as though the PAHs are actually making sense as a fire proxy more-so than MAs- lower values during the times of increased ISM rainfall, higher values during the times of decreased ISM rainfall. Furthermore, the "noise" in the PAH record looks like millennial scale fluctuations in the fire activity, which you aren't capturing in the MA records. I would suggest more discussion on the PAHs as potentially tracking fire activity, instead of just writing them off as being biogenic in nature. There are many different ratios of PAHs that studies have shown to prove useful in determining PAH source (i.e. biomass burning vs fossil fuel burning, biogenic vs burning, etc: : :). Possibly look into some of these ratios as well to see if you can determine a ratio that is suitable for developing your story. Some papers that use ratios include: Denis et al. (2012). PAHs in lake sediments record historic fire events: Validation using HPLC-fluorescence detection. Org Geochem. Miller et al. (2017). Local and Regional Wildfire Activity in Central Maine (USA) during the past 900 years. Journal of Paleolimnology Yunker et al. (2002).

Sources and significance of alkane and PAH hydrocarbons in Canadian arctic rivers. Estuar Coast Shelf Sci

A: We agree with all your comments. We checked the papers you suggested and used them for our interpretations. As you suggested in a comment below, we also checked the target ions for each compounds and repeated the GC analysis for PAHs fractions, with new interesting results. Therefore, we were able to calculate some diagnostic ratios, such as Ant/(Ant+Phe), IP/(IP+Bghi) and FluA/(FluA+Pyr). For example, in the new version of the paper, the description of the diagnostic ratios sounds as: "PAH diagnostic ratios used in this study are Ant/(Ant+Phe), IP/(IP+Bghi) and FluA/(FluA+Pyr). Ant/(Ant+Phe) values generally discriminate between petroleum (< 0.10) and combustion (> 0.10) sources; IP/(IP+Bghi) distinguishes between different combustion sources, with values > 0.5 for grass, wood or coal combustion, values between 0.2 and 0.5 for liquid fossil fuel combustion and values < 0.2 for petroleum sources; FluA/(FluA+Pyr) is used to define the transition point (0.5) between petroleum and combustion ( Denis et al., 2012; Yunker et al., 2002a; Yunker et al., 2002b; Yunker et al., 2015; Zakir Hossain et al., 2013). In Paru Co these ratios are plotted with absolute error bars (Fig. 2g, 2h, 2i), in order to highlight that the influence of error propagation from the original analysis to the ratio values [MOU1] should be carefully investigated when assigning sources from the ratios. Considering the error bars, the three ratios shows values > 0.10 for Ant/(Ant+Phe), > 0.5 for IP/(IP+Bghi) and > 0.5 for FluA/(FluA+Pyr) for the majority of the analysed samples".

I'd like to note that just through comparing Figures 2 and 6 by eye, it seems like the PAH record tracks the GCD regional composite record fairly well (at least way better than the MA record does). I would advise plotting the PAH curve on figure 6 – that way we can visualize how the PAH record tracks regional fire activity.

A: We performed a deep revision to the paper and, due to the fact that both referees #1 and #2 both commented on the difficulty of interpretation of the charcoal composite record within our Paru Co record, we decided to exclude this part of the work. We totally

removed the confrontation with GCD, that was not improving the data interpretation. Moreover, as we responded to referee #1, an area of 1000 km or more could be too wide to be considered as reference for Paru Co.

Minor Comments: Page 13, line 23-24: "In general, fire history shows a decreasing trend from 8 cal ky BP to the present" – this isn't apparent based on the figures you show. The MAs decrease, but the PAHs steadily increase. Distinguish between the two instead of saying "the Paru Co fire history"

A: We agree with your comment, indeed, in the revised paper, we paid attention on keeping distinguished the fire recorded by PAHs from the MAs one. The sentence to which you refer was completely removed from the discussion, due to the new results that implied a deep rewriting and new interpretative points of view. For example, a new part of the discussion sounds as: "During the early Holocene (10.8-8.5 ky BP), levoglucosan, galactosan, mannosan and 5-6ring-PAHs show similar trends, with a general decreasing pattern and some higher peaks at 10.5-10, 9.2 and 8.5 cal ky BP. Both levoglucosan and PAHs record fires during the middle Holocene between 6.5-4 cal ky BP. PAHs increase during the late Holocene from 3 to 1.3 ky BP, while levoglucosan also contains peaks during this time period (Figure 2). The peaks in higher molecular weight PAHs during the early Holocene (Figure 2d) may be explained by local fires with higher combustion temperatures, due to the fact that the higher number of rings requires greater burning energy (Denis et al., 2012). High percentages of 4 -6 ring PAHs generally suggest the contribution of local high-temperature combustion origins (Yang et al., 2016), where such combustion may be the source of BePyr, the congener with the second highest concentration in Paru Co, but also of IP and Bghi, which are high temperature markers (Zakir Hossain et al., 2013). When fuel sources are uniform, hotter fires (at and above 500 °C) commonly produce high concentrations of BePyr, IP, Bghi (Zakir Hossain et al., 2013; Mcgrath et al., 2003). The lower, but not lacking, presence of 3ring and 4ring-PAHs could be due to the fact that lower molecular weight PAHs are more depleted due to weathering processes (Zakir Hossain et al.,

2013). Their lower concentrations may also be due to the fact that the 3ring and 4ring-PAHs could have travelled farther as they are more volatile than the 5-6 ring PAHs. In addition, the 3ring and 4ring-PAHs may have photochemically degraded in the gas phase after emission to the atmosphere (Wang et al., 2010). Higher molecular weight PAHs are more stable compounds compared to 3-4 rings PAHs. If we assume that low molecular weight PAHs degrade at 500 °C, we have to assume that MAs may also degrade at this temperature, as maximum concentrations occur at burning temperatures centred around 250 °C (Zennaro et al., 2015 and references therein). In the Paru Co record levoglucosan concentrations are higher than PAHs during the early Holocene. Therefore, in order to explain this discrepancy, regional early Holocene fires must have been more frequent than local fires, producing high amounts of MAs, without excluding that atmospheric transport of levoglucosan to Paru Co was more efficient during the early Holocene. Therefore, this high abundance of levoglucosan may also be related to a regional signal, as MAs are capable of travelling hundreds of kilometres (Schüpbach et al., 2015; Zennaro et al., 2014). MAs continue to decrease from 8.5 ky BP to 1.5 ky BP whereas 3, 4, and 5ring-PAHs start increasing before reaching their greatest values between 6.5 and 4 ky BP (Figure 2-a,b,c,d,e,f). [...]".

Page 17, line 7: this should be labeled the MA fire history, not the overall fire history from your record. The PAH fire history shows the opposite of this – with lowest values at 8 cal kyr BP, and then a long term increasing trend. It could be beneficial to include coring location in lake and a bathymetric profile of the lake. Looking at lake bathymetry could give insight or possibly explain some of the trends seen in the data, and could manipulate the age-to-depth model so that it isn'tlinear in reality.

A: Thanks for the comment. As we answered previously, we kept separated MAs and PAHs fire histories in the new version of the paper. Regarding the bathymetric profile, we did not introduced it in the figure, but we added a phrase in section 2 (study area) as following: "The lake's watershed is 2.97 km2 and consists of a sloping glacial valley measuring 0.5 to 2.0 km in length with lateral mountain crests higher than 5000 m asl.

The maximum water depth of the modern lake is 1.2 m, with gently sloping sides, but may tolerate a total water level of about 3 m. A central ephemeral stream channel and a second incised channel drain the lake's watershed and feed Paru Co with runoff. Outflow from the lake drains via a small stream channel located approximately 430 m west of the primary outlet (Bird et al., 2014)", and we also considered data for the Holocene lake level changes in our interpretation (from Bird et al., 2014), plotting them in the new figure 4, together with dD n-alkanes (Bird et al), ACL, Paq, summer insolation at 30° N (Berger and Loutre, 1991).

Please check target ions for each compound – for example, many studies that look at retene have a target ion of 219 instead of 234 (the compound's molecular weight). The mass spectra can be found here: https://webbook.nist.gov/cgi/cbook.cgi?ID=C483658&Units=SI&Mask=2380#IR-Spec. Using 234 may be adequate, but in some cases using non-major ions may "hide" compounds, particularly when running in SIM mode on a GC-MS. In your case, it seems that this may in fact be occurring, since you report retene was undetectable in most/all samples. Given the fact that retene is produced by combustion of coniferous trees, its surprising that retene is not found, given the fact that you mention a coniferous forest near the lake (line 24, page 5).

A: As you suggested, we checked again the target ions and ran again the samples on the GC-MS. Retene was detected and considered in the interpretation of the data, as we stated in the new version of the paper: "MAs continue to decrease from 8.5 ky BP to 1.5 ky BP whereas 3, 4, and 5ring-PAHs start increasing before reaching their greatest values between 6.5 and 4 ky BP (Figure 2-a,b,c,d,e,f). This difference may due to higher percentages of lignin burning (evidenced by retene peaks – Supplement S5) with respect to cellulose burning (represented by MA concentrations). Pollen profiles (Zhao et al) indicate an increased presence of trees between 7 and 3 ky BP, coincident with major peaks of retene (Supplement S5). The combination of low concentrations of the 5-6ring PAHs but abundant FluA, Pyr and BePyr suggests relatively geographically

small , but more frequent wildfires (Zakir Hossain et al., 2013). This is what probably happened near Paru Co between 6.5 and 4 cal ky BP".

The spikes seen in all data at the beginning of the record has me skeptical of whether or not it is a true signal of some climate/environmental variability. Adding in some discussion (1-2 paragraphs) on other, more plausible causes of this (i.e. an event in catchment that was preserved in the sediment record, a coring artifact, etc: : :) could give your arguments more validity throughout the manuscript.

A: We agree with you the fact that the high sedimentation rate found in the deepest part of the core could have been derived by a distortion during the coring process or bioturbation, causing the higher fluxes of biomarkers. Due to this high uncertainty, we decided to discuss our biomarker's dataset only until 10.78 cal ky BP, as we state in our revised paper (P.6 L.12-14): "Since the deepest part of the core shows much higher sedimentation rate that cannot be clearly explained, with the possibility of data distortion, the subsequent description and discussion of the results exclude the samples aging 10.784-10.937 cal ky BP, limiting the dataset interpretation to the period between 1.347 and 10.768 cal ky BP".

Figure Comments: Fig 1 a) the map seems a more complex than is necessary. The surrounding areas may not be as important to this study as the TP, so one option could be zooming in on the study region. One option could be to make it similar to the map in the supplement – that map is simpler and much easier to read, and having a map similar to that could more easily highlight the study areas in this figure. Also, you might want to confirm with google about publishing google map images in academic journals – I'm unsure if there are any special permissions needed from Google, but it could be a good thing to check.

A: In the new version of the paper we added a focus on atmospheric transport to the Tibetan Plateau. That is why in our opinion figure 1(a) is important to understand the continental position of the lake within the neighboring geographic areas. As you

suggested, we have zoomed in on the study region. We had already checked on Google's permission's rules (https://www.google.com/permissions/geoguidelines.html) and the use in journals is allowed citing the sources.

Fig 2) do not overlap a) and b). This makes it seem as though there is a peak in values occurring at 4 cal kyr BP. There are multiple ways to fix this – you can either separate them so they don't overlap, or possibly highlight/box the areas in a) that are being zoomed in on in fig b).

A: All the figures from 2 to 5 have strongly changed. Now figure 2 shows sum of PAHs, 3-4-5 ring PAHs, levoglucosan, mannosan, PAHs diagnostic ratios.

Fig 3) try moving a) and c) y axes over to the right side – that way the axes are not overlapping or too close together.

A: All the figures from 2 to 5 have strongly changed. Now figure 3 is much more readable and alternating axis left/right are used in the figures.

Figs 2 and 5) use the same color between these two plots for similar things. For example, in figure 5 you use a gold line to separate ISM changes, while in figure 2 it is a blue line. Try to stay consistent in color schemes for the reader.

A: All the figures from 2 to 5 have strongly changed. We paid attention on using the same colours to indicate similar things.

Fig 6) needs to be higher resolution. On figures b and c, you can barely see the lines. Making the lines bolded/bigger, as well as saving a high resolution image, would help fix this issue. Furthermore, adding the PAH record, not just the MA record, would be very beneficial, as the PAH and GCD records seem to track eachother.

A: As already stated, the comparison with the GCD was removed, and consequently also figure 6 was deleted.

---

## Author Comment (AC5) · 9 Jul 2018

J. L. Toney (Referee) The manuscript "Fire, vegetation and Holocene climate in the south-eastern Tibetan Plateau: a multi-biomarker reconstruction from Paru Co" uses a suite of biomarkers to assess vegetation and fire change during the Holocene from a sedimentary record of a small lake on the Tibetan Plateau. This study presents original data with potentially interesting new results on fire history that has not been widely studied on the Tibetan Plateau. The manuscript is well written and methodologies are robust. Given that this is a relatively new field of research, mainly the application of fire-related biomarkers to paleoclimate records, there are additional aspects that the

authors should consider.

A: We are grateful to Dr. Toney for the useful review and the kind suggestions. We improved the paper according to the suggestions and we incorporated most of the indicated references. We reply to the specific comments below.

PAHs: For instance, although they suggest that there are only a few studies of PAHs as tracers of biomass burning and only cite two (Page 3, Line 12), there are others out there that may help with their interpretations, for instance: Page et al. 1999, Marine Pollution Bulletin; Yunker et al. 2002, Organic Geochemistry; Denis et al. 2012, Organic Geochemistry; Yan et al. 2014, Environmental Toxicity and Chemistry; Yunker et al. 2015, Organic Geochemistry; and Denis et al. 2017, Organic Geochemistry. In particular, not all PAHs result from biomass burning, so using the sum of PAHs, for example, may not be as useful as targeting the pyrogenic PAHs (examples in Page et al. - fluoranthene, phenanthrene, benzo(e)pyrene). Denis et al. 2012 suggest that there are differences in high molecular weight PAHs representing the intensity of the fire (also see McGrath et al. 2003, Journal of Analytical and Applied Pyrolysis), whereas, the low molecular weight PAHs more consistently record local fire events. These considerations may or may not be applicable, but could be tested without acquiring more data. This analysis may help to resolve differences in between the MAs and the PAHs.

A: Thanks for the useful guideline. We incorporated your changes in the sentence (page 3 lines 5-10 of the new version) as following: "PAHs are semi-volatile, persistent, and ubiquitous in the environment with multiple possible sources, and therefore commonly detected in soil, air, and water (Abdel-shafy and Mansour, 2016; Johnsen et al., 2005). The investigation on PAHs as tracers of biomass burning in past climate archives such as sediments (Jiang et al., 1998) and ice (Gabrieli et al., 2010) increased in the last decades (Yan et al., 2014; Page et al., 1999)". Following also the indication from referee#3, we reanalysed all the PAHs fractions, obtaining new interesting results which also include the application of some diagnostic ratios: Ant/(Ant+Phe), IP/(IP+Bghi) and FluA/(FluA+Pyr). We also looked at the signal from different group

of PAHs according to their molecular weight. The paper went, therefore, through a deep revision in sections 4 and 5 (Results and Discussion). For example, a new paragraph describing PAHs results sounds as: "The lowest PAH value is 0.2 ng g-1 of benzo[b]fluoranthene (BbFl) while the highest PAH concentration is 310.3 ng g-1, of naphthalene (Naph). Phenanthrene (Phe), benzo[e]pyrene (BePyr) and Naph respectively represent 20.9%, 18.9% and 17.5% of the total PAH signal in Paru Co. The total sum of PAHs () shows higher valuesin the middle Holocene, with major peaks at 6.3, 5.8, 5.2, 4.8, 3.9 and 3.5-3.3 cal ky BP. The general trend shows increases from 2.2 to 1.3 cal ky BP. The molecular weight and/ or number of aromatic rings of PAHs allows investigating the influence of different PAH types through time. The group of 3ring-PAHs includes Phe, anthracene (Ant) and fluoranthene (FluA), demonstrating a similar pattern to the . The group of 4ring-PAHs encompasses pyrene (Pyr), benzo[a]anthracene (BaAnt), chrysene (Chr), retene (Ret), benzo[b]fluoranthene (BbFl) and benzo[k]fluoranthene (Bkfl), which also has higher values during the middle Holocene and then an increasing trend towards 1.3 cal ky BP. The group of 5-6ring-PAHs is composed by benzo[a]pyrene (BaPyr), BePyr, benzo[ghi]perylene (Bghi), Indeno[1,2,3,-c,d]pyrene (IP) and dibenzo[a,h]anthracene (DBahAnt), with a more noisy trend and dissimilar behaviour from the rest of the groups. 5-6ring-PAHs are high in the early Holocene, peaking at 10.3-9.9 cal ky BP, and then have separate high concentrations at 9.3, 8.6, 7.2, 5.2, 3.9, 3.5, 2 and 1.3 cal ky BP". Another paragraph in which we discuss the PAHs ratios sounds as: "The diagnostic ratios and associated error propagation (Figure 2g, 2h, 2i) do not allow quantitatively assigning PAH sources. IP/(IP+Bghi) contains values above the 0.5 threshold for combustion of wood, wood soot and/or grasses, creosote, as well as almost all wood and coal combustion aerosols and bush fire (Yunker et al., 2002b). The FluA/(FluA+Pyr) ratio, with values above 0.5 for the majority of the samples, indicates the combustion of grass, wood or coal, although this threshold is not definitive (Yunker et al., 2002b). The Ant/(Ant+Phe) ratio with values > 0.10 is generally related to pyrogenic PAH sources, but overlapping values between petroleum and combustion sources are reported (Yunker et al., 2002b).

In Paru Co, when including the error propagation, the majority of samples show values of Ant/(Ant+Phe) > 0.10. Due to the improbability that petroleum sources were burned near Paru Co during the geological time period covered by the analysed core, the obtained values for the ratio Ant/(Ant+Phe) must be related to vegetation combustion. In general, Ant undergoes more rapid photochemical reaction in the atmosphere than Phe. In contrast, FluA/Pyr and IP/Bghi isomer pairs degrade at comparable rates and the original composition information is preserved during atmospheric transport (Yunker et al., 2002b) suggesting that their ratios may be more reliable compared to the Ant/(Ant+Phe) ratio. Given these considerations, we confirm that diagnostic ratios are important tools for the source assignment but cannot be completely trusted due to overlapping values and error propagation that may hinder the correct allocation of the signal origin. However, PAHs in Paru Co can function as pyrogenic markers as we did not find any evidence of other sources (e.g. volcanic eruptions, anthropogenic emissions)".

Finally, with respect to the PAHs (Page 13, Line 28), if there is a change in the biogenic/ diagenic signal of the PAHs, then it would likely manifest specifically in the PAHs like perylene - it would be worth having a look at how the individual PAH profiles change when this signal becomes prominent. Degradation, if it is of the overall organic material should also manifest in changes in the carbon preference index (CPI) values, but these data are not plotted. If a low CPI is seen during times of highly charred material, this index could help support the argument made on Page 16, Line 3.

A: As we answered to the previous point, we looked at the at the signal from different group of PAHs according to their molecular weight. Due to the fact that the new interpretation addressed PAHs as pyrogenic derived, we no more discussed the biogenic/diagenic sources. Regarding CPI, we calculated the ratio in range of chain length of 21-33 and we compared it to the fire signals, in the new figure 3. Our interpretation of these results in the new discussion section 5.2 "combustion sources" sounds as: "In order to obtain more information from the burning conditions, we compared CPI values to L/M and PAHs. Considering that PAHs and n-alkanes are both local indicators, variations in CPI corresponding to spikes in local fire markers may link combustion and vegetation types demonstrated by n-alkane abundances. While no correlation exists between PAHs and CPI, the CPI and L/M have a slight positive correlation (r = 0.31, p-value = 0.03). Medeiros and Simoneit (2008) found that the n-alkanes in green vegetation smoke contained distributions ranging from C23 to C35, with strong odd-to-even carbon number predominance evidenced by CPI ranging from 9 to 58. MAs are better at recording smouldering fires than are PAHs, which may in part explain the similarity between MA and CPI variability through time. The Paru Co CPI values peak around 10 cal ky BP, in the period between 7.8 and 3.5 cal ky BP, and at 2.3 cal ky BP, with values up to 41.2, similar to the peak distributions of L/M. Another argument for relationship between CPI and MAs fire is the fact that lower temperature fires (MAs) essentially steam-distil the vascular plant lipids into the smoke, while high-temperature fires (PAHs) can result in decrease of the CPI (Schefuss et al., 2003 and references therein). In addition, the distance from the vegetation to the sediments may influence the CPI record as plants that are in or near the water pools contain shorter carbon chains, whereas more distant plants have higher CPI values (García-Alix et al., 2017). From these considerations can be assumed that, when CPI parallels L/M, fire from the surrounding areas, and not only near the lake catchment, could have been recorded".

n-Alkanes: It is worth applying some caution in the use of the Paq from Ficken et al. 2000, which was derived in from Mt. Kenya in Africa. The organic geochemistry community is finding that a site-specific approach may be needed and while the assertions about long-chain and short-chain n-alkanes generally hold true, in some environments the relationship is slightly more complex. For example, in Garcia-Alix et al. 2018, Scientific Reports, the supplemental information shows how this index and the ACL vary with distance from water source. Because grasses are prominent during more humid conditions in the arid Sierra Nevada region, the C31 shows aquatic rather than terrestrial-type vegetation changes. This may apply to similar high-elevation sites on the Tibetan Plateau and should be discussed. It is not a fault of the authors, just a

really recent paper that might change the interpretations made here. This could help explain why the n-alkanes are showing a different pattern of change than the MAs and the grass/wood prevalence of pollen data (Page 16, Line 27). Overall, this is a very interesting and well thought out study, but further analysis given the above comments could help with discussion.

A: Thanks for your comments. We checked both Garcia-Alix et al. 2017, Scientific Reports and Garcia-Alix et al. 2018, Scientific Data, finding your useful consideration, as well as other papers. We focused our discussion on Paq and ACL in relation to lake level changes (in the new figure 4) and we found increasing ACL (and diminished Paq) when lake levels are lower and vice versa. Part of the new paper in which we explain this fact sounds as: "Fluctuations in lake levels (Fig. 4d) can be associated with fluctuations in Paq, suggesting a general relationship between higher lake levels and the prevalence of submerged plants between 10 and 5 cal ky BP. The opposite situation occurs between 5 and 1.3 cal ky BP, when a decreasing trend in lake level corresponds to diminishing Paq values. ACL confirms this trend with where the majority of values near 25 occur during higher lake levels (10-5 cal ky BP) and majority of values around 27 occur from 5 to 1.3 cal ky BP. The se high lake levels are consistent with wet conditions from a more intense ISM prevailing until $\sim$ 6 cal ky BP, as evidenced by dD wax and pollen records (Figures 3 and 4)." Thanks again for your comments that really improved the quality of our paper.

---

## Author Comment (AC6) · 10 Jul 2018

Please find below the captions and the figures from 2 to 5 for the new version of the paper.

Captions:

Figure 2: (a) Sum of PAHs concentrations; (b) Sum of 3rings PAHs concentrations (Phe, Ant, FluA); (c) Sum of 4rings PAHs concentrations (Pyr, BaAnt, Chr, Ret, BbFl, Bkfl); (d) Sum of 5-6rings PAHs concentrations (BaPyr, BePyr, Bghi, IP, DBa-hAnt). Data points (black) with absolute error range (grey), LOWESS smoothing with

SPAN parameter 0.2 (red), b-spline interpolation (cyano). (e) Levoglucosan concentration; (f) Mannosan concentration. Data points (black) with absolute error range (grey), LOWESS smoothing with SPAN parameter 0.2 (purple), b-spline interpolation (dark blue). Pink bar indicates the early Holocene period where levoglucosan and 5rings PAHs show high concentrations. (g) Ant/(Ant+Phe); (h) IP/(IP+Bghi); (i) FluA/(FluA+Pyr). Ratios values (black points) with absolute error bars (grey) and diagnostic thresholds (red dashed lines).

Figure 3: (a) L/M ratio values (black points) with absolute error bars (grey); LOWESS smoothing with SPAN parameter 0.2 (light sea gree), b-spline interpolation (blue). (b) CPI ratio values (black points); b-spline interpolation (dark red). (c) Tree pollen (%) from Zhao et al., 2011 - http://apps.neotomadb.org/Explorer/?datasetid=14619. (d) Sum of PAHs concentrations, data points (black) with absolute error range (grey), LOWESS smoothing with SPAN parameter 0.2 (orange), b-spline interpolation (red). Grey bars evidence periods of more intense fire.

Figure 4: (a) $\delta$D wax for C27 and C29 n-alkanes referenced to Vienna Standard Mean Ocean Water scale, data from Bird et al., 2014 - https://www.ncdc.noaa.gov/paleo/study/16399. (b) ACL ratio values (purple points), adjacent-average smoothing with 5 points (black), b-spline interpolation (purple line). (c) Paq ratio values (brown points), adjacent-average smoothing with 5 points (black), b-spline interpolation (brown line). (d) Principal component 1 values (blue) as indicative of lake level changes, adjacent-average smoothing with 40 points (red), data from Bird et al., 2014 - https://www.ncdc.noaa.gov/paleo/study/16399. (e) Summer insolation, data from Berger and Loutre (1991). Pink bars evidence more intense ISM events and/or lake level changes.

Figure 5: (a) Sum of PAHs concentrations, data points (black) with absolute error range (grey), LOWESS smoothing with SPAN parameter 0.2 (red), b-spline interpolation (cyano). (b) lithics (%), data from Bird et al. (2014) - https://www.ncdc.noaa.gov/paleo/study/16399. (c) MAs concentrations, data points

(black) with absolute error range (grey), LOWESS smoothing with SPAN parameter 0.2 (red), b-spline interpolation (cyano). Grey bars indicate less lithics abundances (less ISM) compared to the fire signals.

[Figure]

[Figure]

**Fig. 1.** FIGURE 2

[Figure]

**Fig. 2.** FIGURE 3

[Figure]

**Fig. 3.** FIGURE 4

[Figure]

**Fig. 4.** FIGURE 5